# Integrative multi-omics defines melanoma drug response networks and ARID1A-dependent resistance mechanisms

Charlie George Barker[1,9,10], Sumana Sharma [ID][1,2,10], Ana Mafalda Santos [ID][2], Konstantinos-Stylianos Nikolakopoulos [ID][3], Athanassios D Velentzas [ID][3], Cristina Tormo-Garcia [ID][2], Anushka Sharma [ID][2], Franziska I Völlmy [ID][4], Angeliki Minia[5], Vicky Pliaka[5], Joseph Clarke [ID][2], Maarten Altelaar [ID][4,6], Gavin J Wright [ID][7], Leonidas G Alexopoulos[5,8], Dimitrios J Stravopodis [ID][3] & Evangelia Petsalaki [ID][1✉]

## Abstract

**Resistance to BRAF/MAPK inhibitors is a significant challenge in melanoma treatment, driven by adaptive and acquired mechanisms allowing tumor cells to evade therapy. We explored early signaling responses to BRAF and MAPK inhibition in a BRAFV600E-sensitive melanoma cell line and a drug-resistant ARID1A-knockout (KO) derivative. ARID1A, frequently mutated in melanoma, is linked to resistance and immune evasion. Through an innovative systems biology approach integrating multi-omics datasets, we identified critical resistance mechanisms. We found that ARID1A-KO cells exhibited transcriptional rewiring, sustaining MAPK1/3 and JNK activity post-treatment, suppressing PRKD1 activation, increasing JUN activity, and disrupting PKC dynamics via elevated RTKs (e.g., EGFR, ROS1) and Ephrin receptor activity. ARID1A-KO also reduced HLA-related protein expression and enhanced extra-cellular matrix components, potentially limiting immune infiltration and immunotherapy efficacy. Our multi-omics analysis revealed PRKD1, JUN, and NCK1 as key resistance nodes, offering potential targets for therapeutic strategies to counter resistance in melanoma.**

**Keywords** Melanoma; Drug Resistance; Data Integration; Network Analysis; ARID1A
**Subject Categories** Cancer; Computational Biology

## Introduction

Melanoma is an aggressive skin cancer originating from melanocytes, largely driven by aberrant signaling in the mitogen-activated protein kinase (MAPK) pathway. Nearly 40–50% of melanomas harbor mutations in BRAF, a key MAPK kinase, with BRAFV600E being the most common (~80% of BRAF mutations) (Cancer Genome Atlas Network, 2015). NRAS mutations, affecting ~30% of melanomas, also drive MAPK activation. Mutated BRAF constitutively activates MEK1/2, promoting unchecked cell proliferation. Melanoma cells harboring mutated BRAF typically exhibit a dependency on the BRAF protein and the components of the MAPK pathway (Poulikakos and Rosen, 2011). While BRAF inhibitors initially show strong efficacy, resistance often develops via MAPK reactivation (Anforth et al, 2013). To counteract this, combination therapies with MEK inhibitors (e.g., trametinib) enhance progression-free survival (Nagel et al, 2016), forming the standard treatment for BRAF-mutant melanoma (Flaherty et al, 2012).

Despite its effectiveness, combination treatment with BRAF and MEK inhibitors often leads to short-lived responses, with ~50% of patients relapsing within 6–7 months (Kozar et al, 2019; Sosman et al, 2012). Resistance occurs in ~80% of cases through genetic and epigenetic changes that reactivate the MAPK pathway (Johannessen et al, 2010; Gruosso et al, 2015; Stark et al, 2015; Doudican and Orlow, 2017; Lu et al, 2017; Dietrich et al, 2018; Kozar et al, 2019; Fröhlich et al, 2023). Additional mechanisms include PI3K-mTOR pathway activation via PTEN loss or tumor microenvironment factors (Zuo et al, 2018; Kodet et al, 2018; Ahmed and Haass, 2018). Resistance can also arise without new mutations through reversible adaptive responses (Shaffer et al, 2017; Fallahi-Sichani et al, 2015), with persister cells sustaining growth despite BRAF inhibition(Lito

[1]European Molecular Biology Laboratory, European Bioinformatics Institute (EMBL-EBI), Wellcome Genome Campus, Hinxton, Cambridgeshire CB10 1SD, UK. [2]Radcliffe Department of Clinical Medicine and Medical Research Council, Translational Immune Discovery Unit, Weatherall Institute of Molecular Medicine, University of Oxford, Oxford, UK. [3]Section of Cell Biology and Biophysics, Department of Biology, School of Science, National and Kapodistrian University of Athens (NKUA), Athens 15701, Greece. [4]Biomolecular Mass Spectrometry and Proteomics, Center for Biomolecular Research and Utrecht Institute for Pharmaceutical Sciences, Utrecht University, Utrecht, The Netherlands. [5]Protavio, Demokritos Technology Park, Building 27, Patriarchou Grigoriou & Neapoleos 27, Aghia Paraskevi, Attiki 15341, Greece. [6]Netherlands Proteomics Center, Padualaan 8, 3584 CH Utrecht, The Netherlands Mass Spectrometry and Proteomics Facility, The Netherlands Cancer Institute, Amsterdam 1066 CX, The Netherlands. [7]Department of Biology, Hull York Medical School, York Biomedical Research Institute, University of York, York, UK. [8]Department of Mechanical Engineering, National Technical University of Athens, Athens, Greece. [9]Present address: University College London Cancer Institute, London WC1E 6DD, UK. [10]These authors contributed equally: Charlie George Barker, Sumana Sharma. ✉E-mail: petsalaki@ebi.ac.uk

et al, 2012). The link between adaptive changes and long-term resistance remains unclear, but impaired DNA replication and repair may contribute (Shaffer et al, 2017; Schuh et al, 2020). A deeper understanding of both how intracellular signaling is immediately rewired upon perturbation and how more stable resistance is achieved over the long term is crucial for developing more durable therapies

Immunotherapy offers an alternative to targeted therapy by blocking inhibitory receptors, enabling immune cells to attack cancer (He and Xu, 2020). In melanoma, checkpoint inhibitors, particularly anti-programmed cell-death protein 1 (PD-1) and anti-cytotoxic T-lymphocyte-associated antigen 4 (CTLA-4), provide durable responses but benefit only 40–50% of patients (Larkin et al, 2019). As a result, both targeted therapy and immunotherapy are recommended as first-line treatments for metastatic melanoma (Larkin et al, 2019; Michielin et al, 2019).

Melanoma targeted therapy relies on tumor dependency on mutated pathways, driven by hotspot mutations in genes such as BRAF, NRAS, KIT, and GNAQ. Unlike these, the AT-Rich Interaction Domain 1A protein (ARID1A), mutated in ~11.5% (Kadoch et al, 2013) of melanoma cases, lacks a hotspot mutation pattern, a feature typical of tumor suppressors (Yang et al, 2015). As a key SWI/SNF complex component, ARID1A influences chromatin remodeling and tumor epigenetics. Its mutations correlate with increased programmed cell death-ligand 1 (PD-L1) expression, high tumor mutational burden, reduced immune infiltration, and impaired mismatch repair (Wu et al, 2014; Shen et al, 2018; Okamura et al, 2020; Li et al, 2020). Genome-wide loss-of-function CRISPR screens also link ARID1A loss to resistance against BRAF/MEK inhibitors (Shalem et al, 2014; Doench et al, 2016). Despite these significant findings, the clinical significance of ARID1A mutations, particularly in the context of melanoma, remains ambiguous. As ARID1A has been implicated not only in resistance to targeted therapies for melanoma but also in modulating the therapeutic responses to immune checkpoint blockade, its study lends to better understanding of the interplay between these mechanisms in developing resistance.

Here, we present an integrative multi-omics study to compare the response of drug-sensitive ARID1A-WT (wild-type) vs resistant ARID1A-KO (knockout) melanoma cell lines to both single and combination drug perturbation. We devised a computational strategy that combined data integration using multi-omics factor analysis (MOFA (Argelaguet et al, 2018)), with a network propagation-based method, phuEGO (Giudice et al, 2024), to extract the predominantly affected signaling networks post-short-term treatment of both resistant and sensitive melanoma cell lines with BRAF inhibitors alone or in combination. This allowed us to interpret the data in a unified framework, providing new insights into the signaling processes activated in response to single/combination drug treatment of both sensitive and resistant melanoma cells.

# Results

## Multi-omics data integration identifies molecular signatures associated with drug response and ARID1A KO

ARID1A was previously identified in genome-wide screens as a resistance factor for MAPK (selumetinib) and BRAF (vemurafenib)

inhibitors (Shalem et al, 2014; Doench et al, 2016). To assess whether ARID1A acts broadly across the MAPK pathway rather than mediating resistance specific to a single drug, we conducted two additional screens in A375 cells using a different MEK inhibitor, trametinib. These screens again identified ARID1A, along with established resistance genes (e.g., NF1/2, KIRREL, MED12, TAF5/6L), further supporting its role in BRAFV600E melanoma resistance. (Fig. 1A; Dataset EV1). Targeting ARID1A or MED12, a previously characterized gene responsible for drug resistance to vemurafenib, with sgRNAs, followed by trametinib treatment, led to sustained pMEK and pERK phosphorylation, indicating non-responsiveness to MAPK inhibition (Fig. 1B).

To investigate signaling and gene regulation differences in drug-sensitive vs. resistant ARID1A-knockout (ARID1A-KO) cells, we used early-passage ARID1A-WT A375 parental and matched ARID1A-KO lines (Fig. EV1A,B), which showed increased resistance to vemurafenib and trametinib (Fig. EV1C). To characterize resistance-associated network rewiring, we collected multi-omics data (proteomics, phosphoproteomics, and transcriptomics) from parental (ARID1A-WT) and ARID1A-KO cells, with or without drug treatment, for 6 h ("Methods"; Fig. 1C). This time point was chosen as it provided a steady-state measurement, with MAPK inhibitor effects on phosphorylation suppression confirmed via a pilot 17-plex Luminex assay (Fig. EV1D).

From the mass spectrometry, we quantified 8139 proteins and 3207 phosphosites after integration of our different experimental runs ('Methods"; Datasets EV2 and EV3). Using RNAseq, we quantified the transcription of 14,376 genes (Dataset EV4). All datasets were reproducible (Appendix Fig. S1A–C). Among the 715 proteins, 372 phosphopeptides and 7557 genes that were (significantly) differentially abundant (FDR adjusted $P$ value < 0.01) between ARID1A-WT vs KO experiments (Fig. EV2; Datasets EV2–4), only 13 were common to all datasets demonstrating the orthogonal nature of the different omics layers (Fig. EV2B).

We applied multi-omics factor analysis (MOFA (Argelaguet et al, 2018)) to identify latent factors driving variation across all data modalities in an unbiased manner (Fig. 1C). This revealed three key factors: Factors 1 and 2 capturing adaptive responses to drug treatments and Factor 3 reflecting sustained resistance due to ARID1A loss (Fig. 1D). The low-dimensional data representation showed minimal differences between single-drug treatments (vemurafenib and trametinib) at the omics level compared to the combination treatment (Appendix Fig. S1A; Dataset EV5). Variance analysis showed that drug-associated factors account for most of the variation in protein, phosphosite, and mRNA levels (Fig. EV3). Factor 2, linked to combination therapy, has no impact on the transcriptome (0%) but explains significant variance in phosphoproteomics (32.5%) and proteomics (28.5%). Factor 3, associated with ARID1A KO, mainly affects the transcriptome (12.7%) and proteome (4.3%; Fig. EV3).

To understand the functional implications of the signaling processes represented by the factors identified above, we sought to place the identified genes within the context of their functional environment, i.e., their interaction networks. To this end, we adapted phuEGO (Giudice et al, 2024), a network propagation-based method, to extract active signatures from phosphoproteomic datasets ("Methods"), for use with the factor loadings taken from MOFA. PhuEGO combines network propagation with ego network decomposition, allowing the identification of small networks that comprise the most functionally and topologically similar nodes to

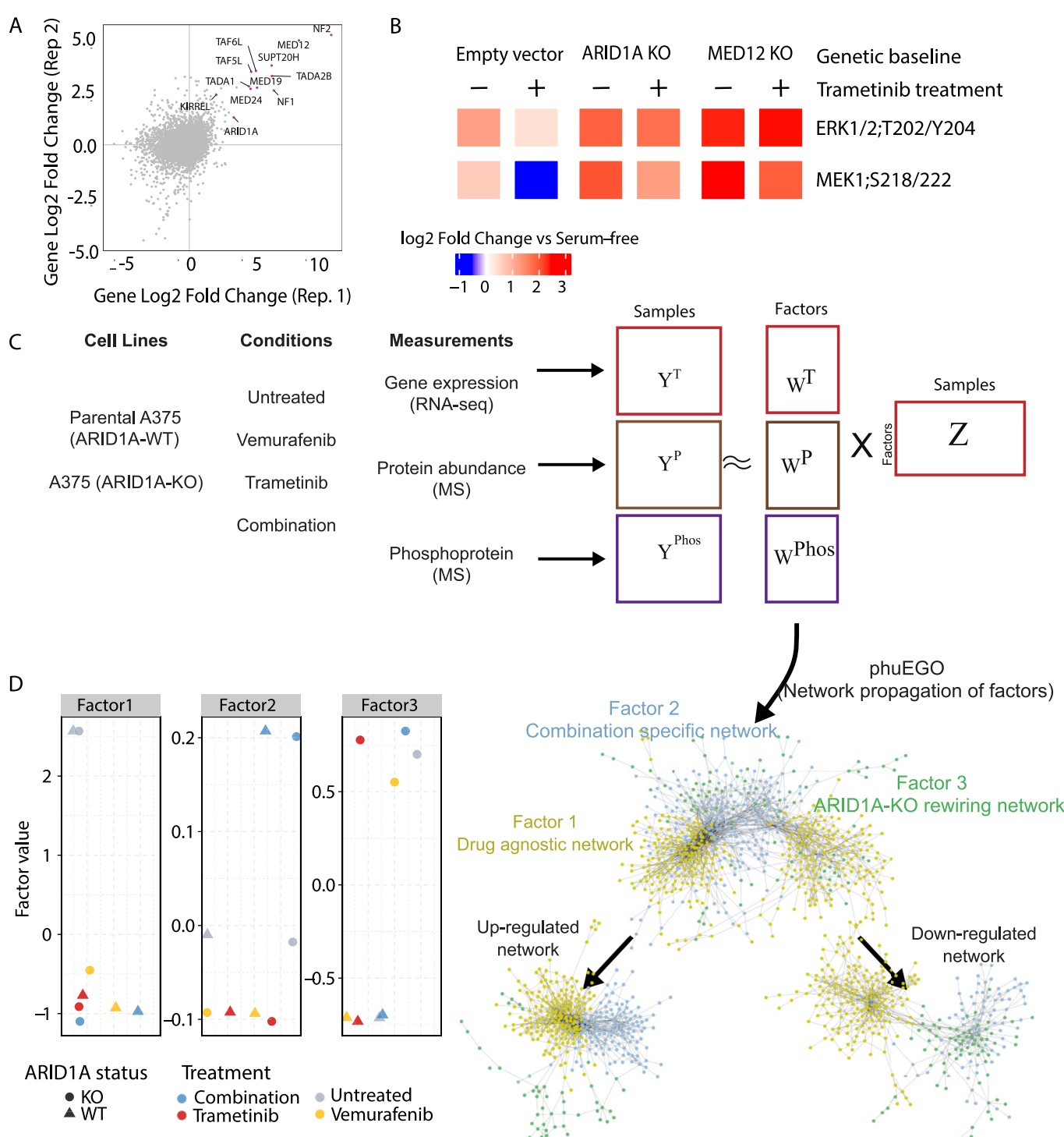

**Figure 1. ARID1A KO in A375 melanoma renders cell lines unresponsive to MEK inhibition.**

(A) Pooled genome-wide CRISPR/Cas9 screens of A375 cells treated with trametinib (MEKi) showing log2 fold change of viability of different knockouts. (B) Luminex assay showing the abundance of phosphosites (MEK and ERK) in different A375 genetic contexts after treatment with trametinib. The color shows log 2 fold change in EGFR-stimulated A375 versus serum-free. Data represent biological duplicates, generated using two independent gRNAs targeting the indicated gene. (C) Schematic presentation of the method employed to integrate and reconstruct signaling networks from melanoma multi-omics data. Up and down indicate positive and negative weights in the factors. (D) Factor weight loadings (*y* axis) for the different samples (colors for drug treatment and shapes for genetic conditions). Source data are available online for this figure.

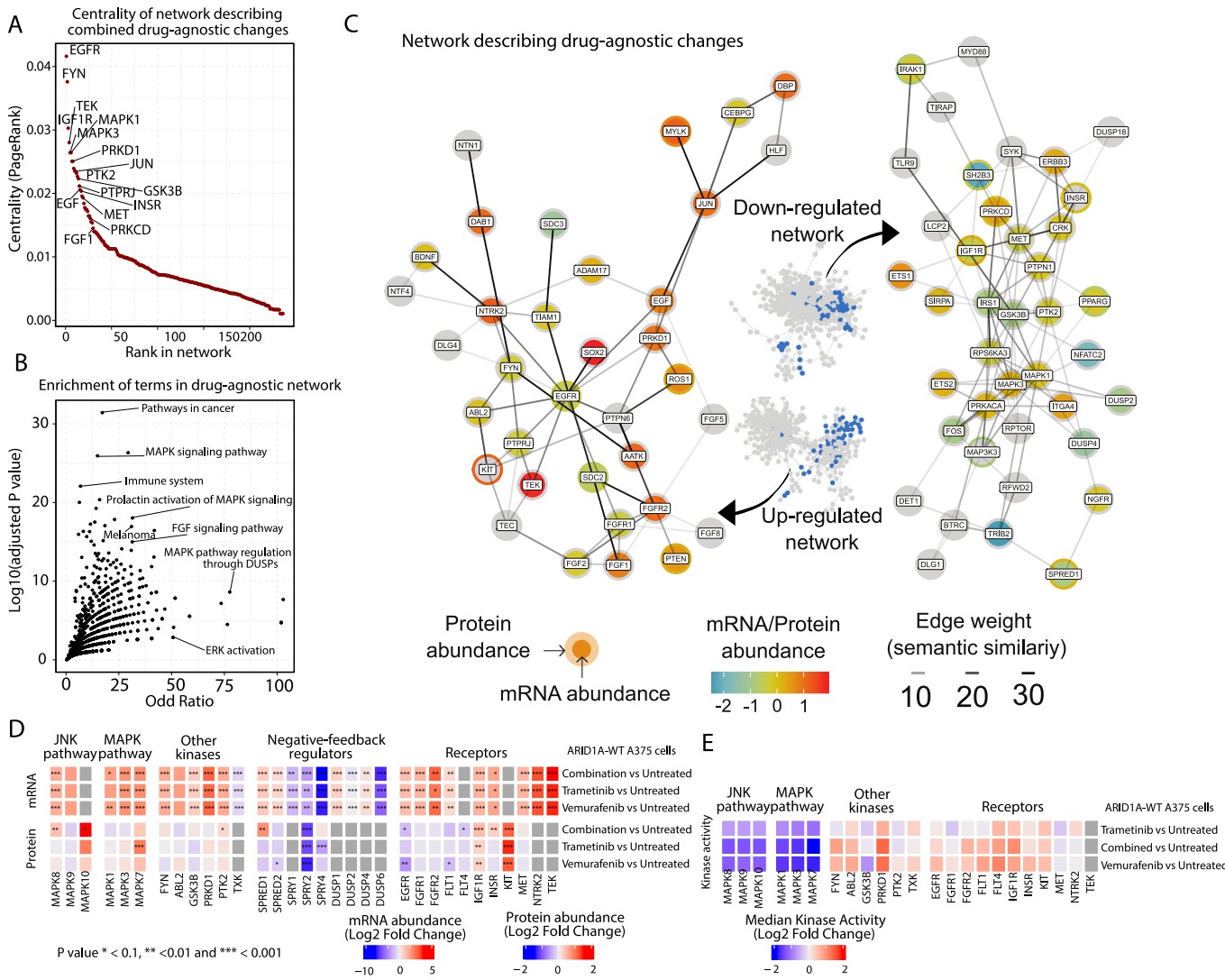

**Figure 2. Overview of molecular signature associated with drug response regardless of type of drug (Factor 1).**

(A) Nodes ordered (x axis) by their PageRank centrality (y axis) in the phuEGO-derived network that are associated with drug-agnostic responses. PageRank centrality reflects the relative influence of each node within the network based on connectivity and neighborhood importance. (B) Processes significantly (y axis) enriched (x axis) in the drug-agnostic phuEGO-derived network. (C) Subset of drug-agnostic phuEGO-derived network highlighting optimized to the 50 most central nodes and their interactions. Semantic similarity indicates the functional similarity between two nodes, as defined by the similarity of their Gene Ontology (GO) terms. The presented mapped protein and transcriptomic factor loadings from MOFA, equivalent to their abundance change in the condition of interest (red increasing after drug treatment and blue decreasing). (D) Heatmap demonstrating the changes in proteomics and transcriptomics abundances for negative regulators of the RTK/MAPK pathway (columns) in different drug conditions in ARID1A-WT A375 cells (rows). (E) Heatmap demonstrating the changes in functional kinomics measurements in ARID1A-WT cells for multiple kinases (columns) found as regulated or relevant in the drug response network for different drug treatments (rows). For both heatmaps, statistical significance was determined using a two-sided moderated t test (three samples per condition, limma) with Benjamini–Hochberg FDR correction. Source data are available online for this figure.

the input ones. This allowed us to generate minimal networks from the factor-specific loadings covering the proteins driving the differences between drug responses and the ARID1A KO, as a function of factor weights (Fig. 1C). This was performed on positive and negative weights and then merged to produce three networks. The resulting positive- and negative-weight networks represent signaling modules that are respectively upregulated or downregulated in association with each factor, thereby capturing opposing functional programs linked to drug response or ARID1A loss.

## Drug-agnostic changes associated with treatment involve negative feedback of RTKs and MAPK

As mentioned above, Factor 1 captures treatment-independent changes, i.e., occurring regardless of BRAF, MEK1/2, or dual inhibition. The network's central nodes include key receptors, kinases, and transcription factors such as EGF, FYN, IGF1R, TEK, MAPK1, and MAPK3, along with PRKD1, JUN, PTK2, and INSR (Fig. 2A; Appendix Fig. S2A). Pathway enrichment analysis of the Factor 1-associated network highlights pathways related to

"Melanoma" and terms related to MAPK signaling, including "DUSP regulation of the MAPK pathway" (Fig. 2B,C).

In agreement with other studies (Gerosa et al, 2020), we observed the negative feedback regulators (e.g., DUSP2 & 4, SPRED1; Fig. 2C) in the downregulated phuEGO-derived network for Factor 1. This agreed with our primary omics data, where we saw the decrease in abundance of the known negative feedback regulators of MAPK (DUSP1/2/4), shown to interact with MAPK3 (ERK1) (Fig. 2D; Appendix Fig. S2B). In both SPRY1/2/4 and SPRED1/2, we detected a decrease in both RNA abundance and protein abundance (Fig. 2D; Appendix Fig. S2B; Datasets EV3 and EV4). The decrease of these negative regulators of RTK and MAPK signaling seems able to relieve downstream inhibition, leading to increased growth factor signaling and ameliorating the effect of MAPK inhibition by vemurafenib or trametinib.

We also found several receptor tyrosine and other kinases in the phuEGO-derived upregulated networks, including EGFR, ROS1, FGFR1 & 2, KIT, FYN, and others, suggesting extensive signaling regulation and rewiring across many pathways upon drug treatment. Protein/transcriptome abundance does not correlate well with kinase activity, and we only found a few phosphosites modulated. Therefore, to study the observed signaling regulation upon drug treatment, we also collected functional kinomics data using the PamChip technology (Krayem et al, 2020), which provides an estimate of multiplex kinase activities in a cell lysate over an array of immobilized target peptides (Appendix Fig. S2C; Dataset EV6). The magnitude of changes in the activity of kinases, in this dataset, correlates well with the centrality of the kinases within our network, with MAPK1 and PRKD1 being the most central kinases quantified in our network (Appendix Fig. S2D). Among them, we found a strong reduction in the MAP kinase activities upon drug treatment (Fig. 2E), even though this was not mirrored by changes in their respective abundance. We also found several other activated kinases, including PRKD1, FYN, and IGFR1, which were shown to be central in our network, as well as ABL2, FLT1, and FGFR2. EGFR presented only a very small increase in activity, which contrasts with its reduction in both transcriptomic and proteomics abundance. Taken together, these results indicate rewiring of RTK-driven signaling following drug perturbation.

## Combination therapy invokes phosphorylation patterns associated with DNA damage repair

Factor 2 illustrates changes that are associated with response to drug combination treatment (Fig. 1D) and are mostly derived from the phosphoproteomics data (Fig. EV3). Drug combination treatments targeting both BRAF and MEK are currently the standard of care for patients with metastatic and non-resectable melanoma, and have shown longer remission of the disease in patients (Robert et al, 2015). Our functional kinomics data indicate that there is practically no difference in the effect on MAP kinases between the combination treatment and single treatment with vemurafenib in our experiment (Fig. 2E). We also did not observe a difference in the killing efficiency of combination treatment compared to mono-treatment for this cell line (Fig. EV1C). We, therefore, decided to zoom in on this factor to shed light on the phosphoproteomic differences observed at this level.

Applying phuEGO to the weights from Factor 2 maps the unique protein networks that are affected by combination therapy (Fig. 1C;

Appendix Fig. S3A). We detect transmembrane receptors, specifically ERBB2, INSR, and EGFR, being central in the network, and surrounded by differentially phosphorylated proteins (Fig. 3A,B; Appendix Fig. S3A). TMEM30A, a protein involved in cell migration, is the second most central node, followed by SRC. AKT1, an alternate growth regulator from MAPK, is also highly central (Fig. 3A,B; Appendix Fig. S3B). The corresponding combination therapy-specific downregulated network is centered around the transcription factor MYC and the DNA damage response protein TP53BP1 (Fig. Appendix Fig. S3B). Also present are the RAF1 and KRAS signal transducers. MAP3K7 (TAK1) is also observed, as it has negative weights due to reduced phosphorylation (Fig. 3B; Appendix Fig. S3C) following combined inhibition of MAP2K1 and BRAF.

Despite phosphoproteomics being the main driver for this factor (Fig. EV3), there was little change in kinase activities, when looking at the functional kinomics data, compared to changes with the single-drug perturbations as mentioned above (Appendix Fig. S3C). Looking at the phosphosites that drive the variance captured by Factor 2, we find several phosphoproteins involved in DNA repair-related functions (Fig. 3C; Dataset EV5). For example, TP53BP1 is heavily phosphorylated on serine residue 1101, while being significantly less phosphorylated after combination therapy on multiple serine residues from residues 222 to 771 (Fig. 3C). TP53BP1 able to promote non-homologous end joining (NHEJ) DNA repair of double-strand DNA breaks (Silverman et al, 2004). Phosphorylation of serine residue 1101 is known to be associated with DNA damage induced by ionizing radiation (Bennetzen et al, 2010; Stokes et al, 2007). RIF1 is known to be a key regulator of TP53BP1 and is also significantly modulated by phosphorylation after combination treatment (Silverman et al, 2004). While there are no functional annotations for these sites, RIF1-S2196, which is downregulated in the combination treatment compared to no or single-drug treatment (Fig. 3C) is very close to S2205, which is known to inhibit the protein's function (Nasa et al, 2020) and is predicted to be phosphorylated by JNK1,3 or P38δ or γ, among other kinases, all of which are downregulated in our functional kinomics dataset compared to no drug treatment (Johnson et al, 2023). SSRP1, which is also known to be involved in DNA repair processes (Keller et al, 2001), shows decreased phosphorylation in S659 (Fig. 3C). ULK1-S556, which is known to be phosphorylated by ATM (Guo et al, 2022) and to be inducing autophagy (Amraei et al, 2020), shows strong upregulation in the combination treatment compared to all other conditions (Fig. 3C). SIRT1-S47 is not one of the phosphosites driving Factor 2, however it is one of the significantly downregulated peptides in combination treatment versus untreated control, and is known to promote epithelial-to-mesenchymal transition (EMT) through autophagic degradation of E-cadherin (Sun et al, 2018). LEO1-S630 is involved in the maintenance of embryonic stem cell pluripotency (Ding et al, 2009; Zhao et al, 2005) and is also downregulated. Finally, we observe phosphoregulation of SRRM2, a component of spliceosome, at multiple phosphosites (Fig. 3C).

Zooming out to look at the processes involved in this network, we performed functional enrichment analysis on both up- and downregulated networks. The upregulated network (Appendix Fig. S3D) showed enrichment in RTK-driven pathways (ERBB, ERBB2, cMET signaling pathways), as well as PI3K/Akt and mTOR signaling ("PI3K events in ERBB2 signaling" and "mTOR signaling pathway"). The downregulated network reflected expected MAPK pathway inhibition and included terms related to DNA damage repair ("ATM-mediated phosphorylation of repair proteins" and

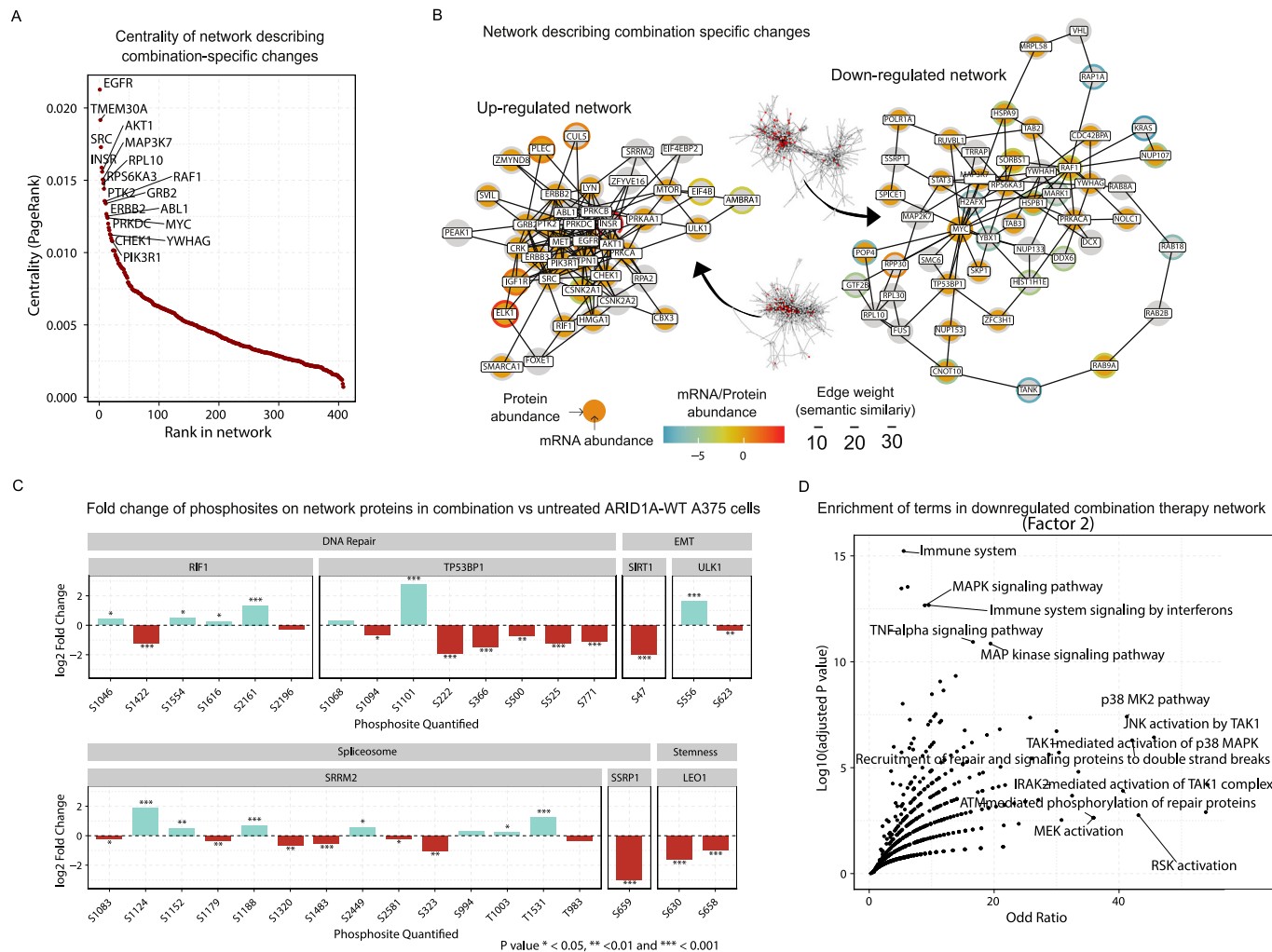

**Figure 3. Overview of molecular signature associated specifically with combination drug response (Factor 2).**

(A) Nodes ordered (x axis) by their PageRank centrality (y axis) in the phuEGO-derived network that are associated with drug-agnostic responses. (B) Subset of combination treatment phuEGO-derived network highlighting the 30 most central nodes and their interactions. Semantic similarity indicates the functional similarity between two nodes, as defined by the similarity of their Gene Ontology (GO) terms. The presented mapped protein and transcriptomic factor loadings from MOFA, equivalent to their abundance change in the condition of interest (Red increasing after combination drug therapy and blue decreasing). (C) Highlighted phosphosite changes in the combination treatment showing the change in abundance (y axis) after treatment with drugs (x axis). Statistical significance was determined using a two-sided moderated t test (three samples per condition, limma) with Benjamini–Hochberg FDR correction. These have been selected to be present in our phuEGO-derived networks and show differential abundance in the combination treatment conditions. (D) Processes significantly (y axis) enriched (x axis) in the downregulated network upon combination treatment using gene-set enrichment analysis (enrichR). Statistical significance of enrichment was assessed using a right-tailed Fisher's exact test, and deviation from expected ranks was quantified using Enrichr's Z-score. The combined score (Z-score × −log10(P value)) was used to rank enriched terms. Source data are available online for this figure.

"Recruitment of repair and signaling proteins to double-strand breaks") and immune signaling (TNF-alpha, interferons, interleukins, prolactin, and growth hormone; Fig. 3D).

## The basal transcriptional state of cells following ARID1A KO influences response to drug treatment

Although ARID1A-KO cells are resistant to MEK/BRAF inhibition (Fig. 4A), their transcriptomic and proteomic drug responses are highly similar to those of parental cells (Figs. 1D and EV4A,B). This is reflected in differential abundance upon drug treatment with a

high correlation between the transcriptomic ($R = 0.91$–$0.93$, $P < 2.2e-16$) and proteomic responses ($R = 0.7$–$0.8$, $P < 2.2e-16$) in the ARID1A-KO vs the ARID1A-WT cells (Fig. EV4C). However, kinase activity assays and phosphoproteomics reveal marked differences in signaling behavior. For example, MAPK1/3 show reduced sensitivity to drug treatment, while Ephrin receptor signaling is elevated in ARID1A-KO A375 cells compared to parental A375 (Fig. 4A).

To explain this observation, we focused on the differences between the two cell lines and found markedly different baseline states. Specifically, ARID1A loss, acting as a transcriptional

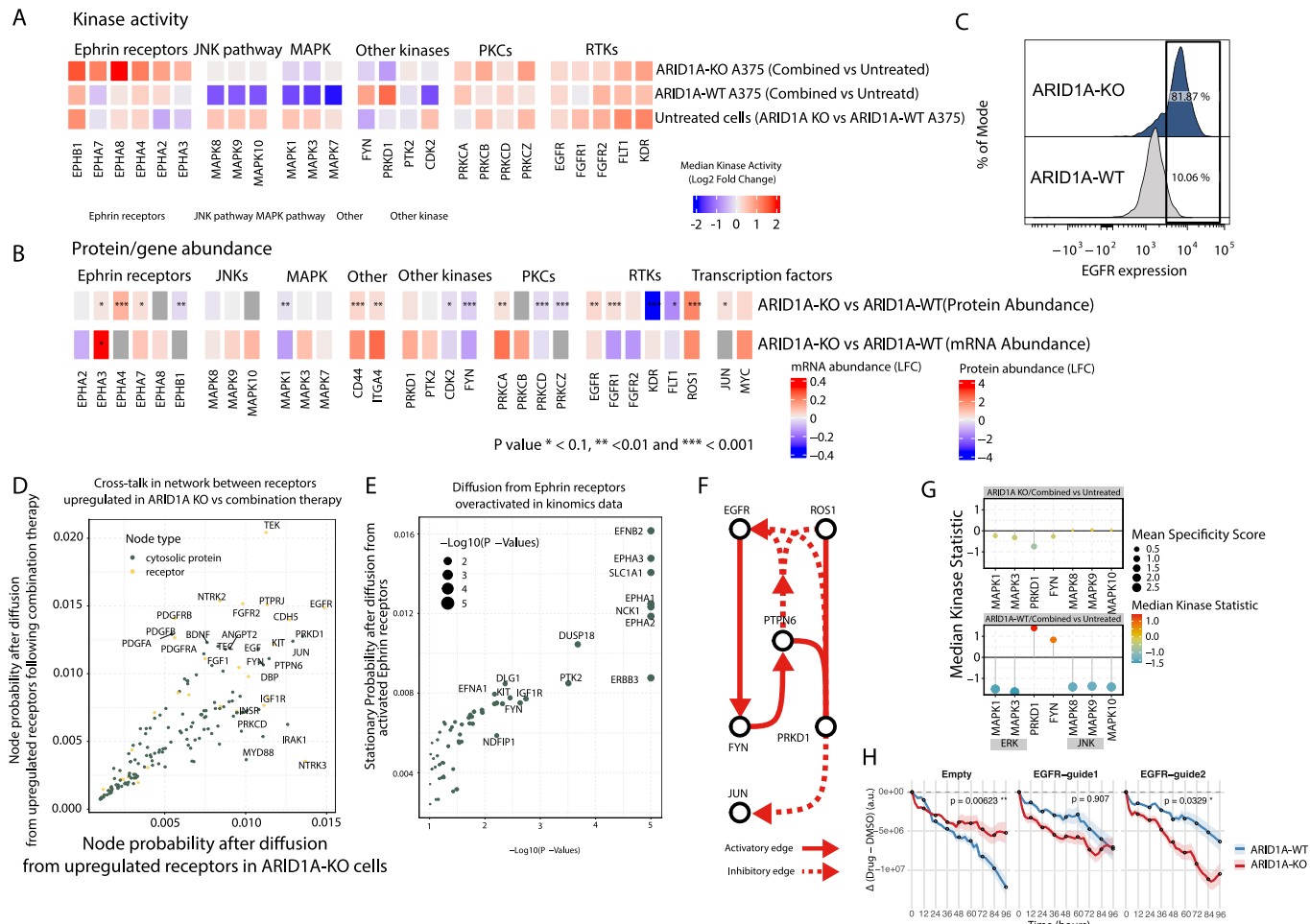

**Figure 4. Summary of changes induced by ARID1A KO and their interaction with drug response.**

(A) Kinase activity of selected kinases on ARID1A-KO and WT cells at a basal level and post-combination drug treatment. (B) Heatmap demonstrating the changes in proteomic and transcriptomic abundances for selected kinases at a baseline state for ARID1A-KO cells compared to the parental A375 cell line. Statistical significance was determined using a two-sided moderated $t$ test (three samples per condition, limma) with Benjamini–Hochberg FDR correction. (C) Expression of EGFR as determined by flow cytometry on the surface of ARID1A-KO or parental A375 ARID1A-WT cell line. (D) Scatter plot showing probability distribution from the network propagation from receptors that are upregulated in response to combination therapy (ARID1A-KO, $y$ axis) and receptors that are upregulated in response to ARID1A KO ($x$ axis). (E) Scatter plot showing probability distribution from the network propagation from Ephrin receptors activated following treatment of combination therapy in ARID1A-KO cells. $X$ axis refers to significance ($-\log10(P)$) and the y axis shows the probability distribution of specific nodes. Statistical significance was assessed using an empirical permutation test (10,000 degree-preserving random graph permutations) comparing the stationary probability of each node to its null distribution. (F) Network showing how paths from receptors converge on JUN via FYN and PRKD1 following network propagation from EGFR and ROS1. (G) Kinase activity assay showing the median kinase activity ($y$ axis) of selected kinases ($x$ axis) following combination drug treatment in parental A375 (ARID1A-WT) cell lines (bottom) and ARID1A-KO A375 cell lines (top). (H) ARID1A-WT and ARID1A-KO A375 cells expressing Cas9 and mOrange were transduced with empty or EGFR-targeting gRNAs and treated with vemurafenib + trametinib. mOrange loss was monitored over 96 h. ARID1A-KO cells showed slower drug-induced death than WT, a phenotype reversed by EGFR targeting ($n = 4$). Statistical significance was assessed using Welch's two-sample $t$ test (two-sided) for the final time point. Source data are available online for this figure.

reprogrammer, reshapes the initial gene and protein expression landscape, thereby rewiring signaling and modifying responses to drug perturbation. Compared with parental cells, ARID1A-KO cells show increased abundance of several receptors affected by drug therapy, including ROS1, ITGA4, EGFR, and CD44 (Fig. 4B). EGFR, in particular, showed a drastic increase of membrane expression in the ARID1A-KO line compared to the parental line, in agreement with impaired receptor endocytosis mechanisms in drug resistant melanoma cells (Gerosa et al, 2020) (Fig. 4C). Oncogenic transcription factors such as JUN and MYC, together with their regulons, are also upregulated (Figs. 4B and EV4D),

suggesting that ARID1A loss primes cells for enhanced oncogenic signaling and drug resistance.

To explore how the functional effect of these baseline changes and how they shape therapy responses, we simulated signaling cross-talk between ARID1A KO-induced receptor changes and drug-induced receptor alterations. To this end, we first integrated networks capturing drug-agnostic response signaling (Factor 1) with ARID1A KO-specific effects (Factor 3), producing a combined map of processes associated with both (Appendix Fig. S4A). Enrichment of pathways within this network revealed strong signals in MAPK signaling, immune system pathways, melanoma,

focal adhesion, and FGF signaling (Appendix Fig. S4B). Next, using a Markov Random Walk approach (Tong et al, 2008), we propagated signals separately from receptors upregulated by ARID1A KO and those induced by drug response, to identify points of cross-talk. This analysis highlighted shared hubs where the two signals converge, including JUN, PRKD1, FYN, PTPN6, and PRKCD (Fig. 4D). We next examined the Ephrin receptor family, which is strongly activated after drug treatment in ARID1A-KO but not in parental (ARID1A-WT) cells (Fig. 4A,E; Appendix Fig. S2C). Network propagation from these receptors identified additional players, such as NCK1, DUSP18, PTK2, FYN, and DAPK, as potential mediators (Fig. 4E).

Comparison of baseline ARID1A-KO protein reprogramming with drug-induced abundance changes further clarified these effects. In ARID1A-KO cells, NCK1 abundance was unresponsive to drug treatment, whereas DUSP18 gained responsiveness. Although FYN and DAPK1 displayed similar treatment responses in both parental and KO cells, their baseline abundances were shifted, thereby changing the effective signaling flux through these nodes (Appendix Fig. S4C).

To identify key routes of this rewired communication, we performed maximum-flow network analysis (see "Methods") to trace the most probable signaling paths toward downstream effectors. This revealed convergent flows from EGFR and ROS1 onto JUN through intermediates including PRKD1, FYN, and PTPN6 (Fig. 4F), consistent with their identification as cross-talk hubs in our Markov propagation analysis (Fig. 4D). Functional kinase assays confirmed that, unlike in parental cells, both FYN and PRKD1 exhibit attenuated drug-induced activation in ARID1A-KO cells (Fig. 4G), demonstrating that these nodes are functionally rewired and no longer respond dynamically to MEK/BRAF inhibition.

To directly test the predicted dependency on upstream receptor signaling, we targeted EGFR using two distinct gRNAs in both ARID1A-KO and parental ARID1A-WT lines and measured cell viability upon combination drug treatment over a time course of 96 h using the IncuCyte imaging system. EGFR loss abolished the drug-resistant phenotype of ARID1A-KO cells and increased cell death beyond levels observed in parental cells (Fig. 4H), confirming that EGFR-driven signaling is a critical determinant of ARID1A-mediated resistance.

Together, these results reveal that ARID1A loss reprograms receptor abundance, membrane signaling organization, and transcriptional activity. Although global transcriptional and proteomic drug responses appear similar between genotypes, the altered baseline signaling topology redirects flow from EGFR and ROS1 through a rewired FYN–PRKD1–JUN axis, explaining the distinct drug responses of ARID1A-KO cells.

### ARID1A KO suppresses HLA proteins in both in vitro and in vivo contexts

To assess how ARID1A KO affects transcriptional regulation in response to drug treatment, we analyzed transcription factor activity upon drug treatment. We observe an increase in basal levels of JUN and MYC activities following ARID1A KO, consistent with our previous findings (Fig. EV4C). Notably, we also observed a significant decrease in the regulons of RFX5, RFXAP, RFXANK, and NFYC, which regulate MHC class-II gene transcription(Sachini and Papamatheakis (2017)) (Fig. 5A). This led to reduced

expression of MHC class-II genes and CD74, while genes repressed by RFX5 (e.g., COL1A2) increased (Fig. 5B).

Although MHC class-II molecules, which present antigens to CD4[+] T-helper cells, are typically expressed in professional antigen-presenting cells (APCs), such as dendritic cells and B cells, A375 cells also express them (Johnson et al, 2016). To confirm our findings, we found significantly reduced surface expression of HLA-DQ and HLA-DR in ARID1A-KO cells (Fig. 5C). In addition, upregulated transcription factors included regulators of extracellular matrix remodeling, such as TWIST and SMAD families (Fig. 5A,B).

Using the TCGA data (https://www.cancer.gov/tcga), we investigated whether these patterns appear in melanoma patients. We stratified 472 patients into ARID1A-affected (mutations or deletions, $n = 80$) and unaffected ($n = 392$) groups ("Methods"; Appendix Fig. S5). ARID1A-affected patients showed significantly lower predicted activity of MHC class-II regulators RFXAP, RFX5, and CIITA (adjusted $P < 0.001$) (Fig. 5D). In addition, SP7, a regulator of collagens and metalloproteins, was significantly upregulated. Both ARID1A-affected patients and KO cells exhibited increased collagen and laminin expression, potentially contributing to ECM stiffness (Huang et al, 2021), along with reduced HLA protein levels (Fig. 5E).

## Discussion

Resistance to MAPK inhibitors remains a major challenge in melanoma, driven by adaptive and acquired mechanisms that enable tumor cells to evade therapy. While most studies focus on either early adaptive responses or acquired resistance, we compared drug-induced signaling changes in both contexts. Using *ARID1A*, a frequently mutated resistance-associated gene, we explored its role in shaping drug response and immune evasion.

To uncover network-level insights, we integrated transcriptomics, proteomics, and phosphoproteomics, addressing limitations of single-metric approaches. Combining integrative matrix factorization (Argelaguet et al, 2018) with a network-centric method that we had previously developed (Giudice et al, 2024), we identified key pathway alterations influenced by drug treatment and ARID1A loss, providing a comprehensive view of resistance mechanisms.

We focused on early cellular responses to drug treatment, selecting a 6-hour time point to capture pathway inhibition and key changes, including reduced DUSPs and other MAPK-negative regulators (Lito et al, 2012). This phosphatase loss disrupts RTK signal termination, increasing signaling activity. Downregulation of *DUSP, SPRY*, and *SPRED* genes confirmed feedback sensitivity, alongside heightened RTK (EGFR, IGFR, FGFR) activity despite minimal gene or protein expression changes.

We used functional kinomics to validate signaling pathways identified through multi-omics integration. Factor 1 network nodes, particularly serine/threonine kinases, showed the most activity changes, including decreased MAPK1/3 activity (the drug target). One striking change was the increased expression and activity of PRKD1, a PKD family kinase activated by RTKs and reactive oxygen species (ROS). PRKD1 suppresses EGF-dependent JNK activation (Bagowski et al, 1999) by forming a complex with JNK (Hurd et al, 2002) and inhibiting c-Jun phosphorylation at serine-63. Frequently mutated and highly expressed in melanoma,

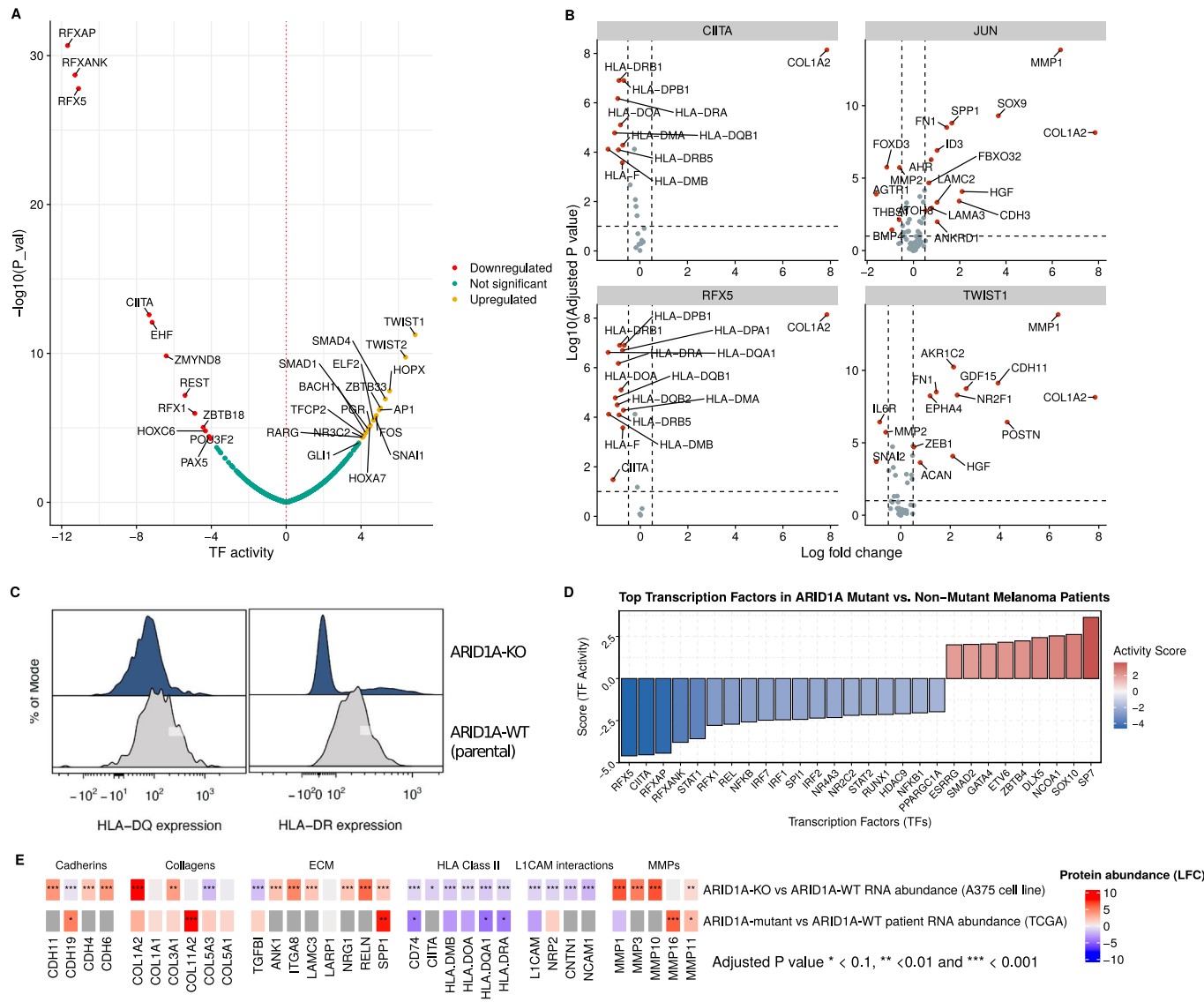

**Figure 5. Overview of molecular signature associated specifically with combination drug response (Factor 2).**

(A) Transcription factor activity changes (x axis) in ARID1A-KO A375 cell line compared to parental ARID1A-WT line. Significance of TF activity was assessed using the t-statistic from a Univariate Linear Model (ULM), where the *t* value of the regression slope quantifies the likelihood that observed gene expression is explained by the TF's target-weight model (3 samples per condition, see "Methods"). (B) Log-fold changes for regulons (x axis) of four key transcriptional regulators (CIITA, JUN, RFX5, and TWIST1) that were identified to have significantly differential (y axis) activity at a basal state for the ARID1A-KO cell line compared to the parental (ARID1A-WT) A375 cell line. Statistical significance was determined using a two-sided moderated *t* test (three samples per condition, limma) with Benjamini–Hochberg FDR correction. (C) Cell surface expression of two MHC class-II antigens (HLA-DQ and HLA-DR) on parental (ARID1A-WT) and ARID1A-KO cell lines. (D) Top transcription factors and regulators (e.g., CIITA) with differential activity (y axis) in ARID1A-mutant versus non-mutant melanoma patients from TCGA. Statistical significance was determined using a univariate linear model (472 samples, see "Methods"). (E) Heatmap demonstrating the changes in transcriptomic abundances for selected genes (columns) at a baseline state for either ARID1A-KO compared to parental (ARID1A-WT) A375 cell line (top) or melanoma samples from ARID1A mutant versus WT patients (bottom). Source data are available online for this figure.

PRKD1 has context-dependent pro- or anti-proliferative effects across multiple cancers (Youssef and Ricort, 2019). Its activation upon drug treatment correlated with reduced JNK phosphorylation, suggesting suppression of the JNK/c-Jun axis, which is linked to apoptosis in mutant-BRAF melanoma (Fallahi-Sichani et al, 2015).

Combination therapy, which targets multiple points in the MAPK pathway, extends efficacy and prevents paradoxical MAPK activation in BRAF wild-type cells (Rasmussen et al, 2024). At the

signaling level, we observed small phosphorylation changes in DNA repair (RIF1) and EMT-related proteins (SIRT1, LEO1), but in our BRAF-mutant model, these changes did not affect cell death rates between mono- and combination therapies. Further research is needed to assess their impact on EMT and melanoma immunogenicity.

While ARID1A-KO and ARID1A-WT parental cells showed similar mRNA and protein responses to drug treatment, signaling

outcomes differed. ARID1A-KO cells maintained MAPK1/3 and JNK activity, indicating resistance. PRKD1 was activated by drug treatment in parental cells but suppressed in ARID1A-KO cells, likely due to transcriptional rewiring. This may have relieved JNK inhibition, increasing JUN activity, a key resistance driver. JUN upregulation is common in BRAF inhibitor-treated melanomas (Fallahi-Sichani et al, 2015), and dual targeting of JUN and BRAF has shown promise in overcoming resistance (Titz et al, 2016).

Our multi-omics analysis identified two key receptor-level signaling differences in ARID1A-KO cells, which we hypothesize contribute to MAPK inhibitor resistance: increased basal RTK expression (EGFR, ROS1) and heightened Ephrin receptor activity post-treatment. Elevated RTK levels can impair receptor endocytosis, leading to chronic activation rather than transient ligand-dependent signaling (Kiyatkin et al, 2020; Lund et al, 1990). Mechanistic modeling has recently suggested that full BRAF inhibition can reactivate MAPK signaling through RTK-stimulated RAS activation (Fröhlich et al, 2023). We propose that increased RTK (EGFR, ROS1) expression and Ephrin receptor activity dominate signaling, altering PKC dynamics and suppressing PRKD1 activation, sustaining high baseline JUN activity—consistent with our observation of increased PKC subunit phosphorylation (PRKCB, PRKCZ) in ARID1A-KO cells.

ARID1A mutations also affect tumor-immune interactions. Its loss disrupts interferon (IFN) signaling, reducing cytotoxic T-cell infiltration and impairing immunotherapy efficacy (Li et al, 2020). Our analysis of ARID1A-KO cells and patient data revealed significant downregulation of IFN-regulated HLA-related proteins (Amaldi et al, 1989). In addition to MHC class II loss, we observed increased ECM component expression, including collagens, laminins, and MMPs—proteins previously linked to resistance in anti-PD1-treated mouse models (Bernardo et al, 2021; Fellermeyer, 2023). These findings suggest that ARID1A KO-driven ECM remodeling may hinder T-cell infiltration and reduce immunotherapy effectiveness.

Consistent with previous studies (Gerosa et al, 2020), we found that receptor-level rewiring plays a key role in resistance. However, we found no evidence that ARID1A perturbs established negative feedback mechanisms (Gerosa et al, 2020). Instead, we observed widespread receptor increases (EGFR, ROS1, FGFR1, ITGA4), suggesting mechanistic redundancy that makes targeting a single protein ineffective, as signaling can be restored through multiple pathways.

Using a network-centric approach, we identified new proteins linked to MAPK resistance, including FYN, PRKD1, and NCK1. NCK1, an adapter protein associated with Ephrin signaling, showed differential drug response: in WT A375 cells, its expression dropped sharply after treatment, whereas in ARID1A-KO cells, it remained unchanged. NCK1 is known to activate both JUN (via JNK (Miyamoto et al, 2004)) and MAPK (via RAS (Yiemwattana et al, 2012)), and in our data, both pathways were unresponsive in ARID1A-KO cells, suggesting a functional interaction consistent with prior literature.

Our study introduces an integrative, graph-based framework that combines multi-omics factor analysis (MOFA) with network-based propagation (phuEGO) to connect latent molecular programs with their mechanistic signaling context. Prior research has demonstrated the utility of MOFA for constructing network representations of multi-omics variation (Dugourd et al, 2021);

however, our approach uniquely extends this concept by integrating factor analysis with diffusion-based network construction across phosphoproteomic, transcriptomic, and proteomic layers. This framework provides three key advantages: (i) it disentangles overlapping sources of variation, e.g., separating drug-induced responses from ARID1A-dependent basal reprogramming; (ii) it mechanistically integrates molecular changes across omics layers, revealing pathway cross-talk such as the EGFR–ROS1–JUN axis mediated by PRKD1 and FYN; and (iii) it prioritizes experimentally testable nodes, enabling targeted validation of predicted dependencies, such as EGFR. Integrating functional kinomics confirmed the accuracy of the inferred network architecture and the predicted signaling vulnerabilities. Together, this combination of factor-based dimensionality reduction and network contextualization provides a powerful and generalizable framework for extracting mechanistic insights from high-dimensional multi-omics data.

In summary, this study provides a systems-level analysis of early cellular responses to MAPK inhibition in melanoma, revealing key adaptive and resistance mechanisms. Using graph-theoretical and multi-omics integration, we show that ARID1A-mediated transcriptional rewiring reshapes signaling, driving resistance to MAPK inhibitors. Notably, ARID1A-KO melanoma cells exhibit distinct alterations in PRKD1, JUN, and receptor tyrosine kinases (EGFR, ROS1), contributing to enhanced resistance.

By integrating functional kinomics with multi-omics data, we identified critical nodes in drug-resistant signaling networks. Our findings highlight receptor activation redundancy, complicating single-target therapies but suggesting alternative combination strategies, such as JUN or immune pathway targeting. These insights deepen our understanding of melanoma resistance and provide a foundation for more effective, genetically tailored therapeutic approaches.

## Methods

**Reagents and tools table**

| Reagent/resource | Reference or source | Identifier or catalog number |
|---|---|---|
| **Experimental models** | | |
| A375 | Synthego | |
| A375-ARID1A KO | Synthego | |
| **Recombinant DNA** | | |
| **Antibodies** | | |
| Anti-EGFR - Alexa 488 | Biolegend | Clone AY13, Cat 352907 |
| Anti-HLA-DQ | Biolegend | Clone HLADQ1, Cat 318105 |
| Anti-HLA-DR | ebiosciences | Clone LN3, Cat 17-9956-41 |
| **Oligonucleotides and other sequence-based reagents** | | |
| EGFR guide 1 forward sequence: CACCGTACACCGTGCCGAACGCAC | Sigma | |
| EGFR guide 1 reverse sequence: AAACGTGCGTTCGGCACGGTGTAC | Sigma | |
| EGFR guide 2 forward sequence: CACCGTAGTTAGATAAGACTGCTA | Sigma | |

| Reagent/resource | Reference or source | Identifier or catalog number |
|---|---|---|
| EGFR guide 2 reverse sequence: AAACTAGCAGTCTTATCTAACTAC | Sigma | |
| **Chemicals, enzymes, and other reagents** | | |
| Trametinib | APExBio | Cat. No.: A3018 |
| Vemurafenib | APExBio | Cat. No.: A3004 |
| Penicillin/Streptomycin/Neomycin | Sigma | Cat. No.: P4083 |
| FBS | Gibco | Cat. No.: 10500-064 |
| DMEM-F12 | Gibco | Cat. No.: 11320033 |
| Lysis buffer | Protavio | Cat. No.: PR-ASSB |
| Protease inhibitor | Roche | Cat. No.: 11873580001 |
| PMSF | Sigma | Cat. No.: 93482 |
| SAPE | MOSS Inc. | SAPE-001 |
| **Software** | | |
| ggplot2 (v3.4.0) | R Core Team | URL: https://ggplot2.tidyverse.org/ |
| ggrepel (v0.9.1) | CRAN | URL: https://cran.r-project.org/package=ggrepel |
| ggpubr (v0.6.0) | CRAN | URL: https://cran.r-project.org/package=ggpubr |
| gridExtra (v2.3) | CRAN | URL: https://cran.r-project.org/package=gridExtra |
| igraph (v1.4.0) | CRAN | URL: https://igraph.org/ |
| EnsDb.Hsapiens.v86 (v2.12.0) | Bioconductor | URL: https://bioconductor.org/packages/EnsDb.Hsapiens.v86 |
| ggVennDiagram (v1.2.0) | CRAN | URL: https://cran.r-project.org/package=ggVennDiagram |
| MOFA2 (v1.4.0) | Bioconductor | URL: https://bioconductor.org/packages/MOFA2 |
| stringr (v1.5.0) | R Core Team | URL: https://stringr.tidyverse.org/ |
| purrr (v1.0.1) | R Core Team | URL: https://purrr.tidyverse.org/ |
| OmnipathR (v2.2.0) | CRAN | URL: https://cran.r-project.org/package=OmnipathR |
| magrittr (v2.0.3) | CRAN | URL: https://cran.r-project.org/package=magrittr |
| dplyr (v1.1.2) | R Core Team | URL: https://dplyr.tidyverse.org/ |
| tidyr (v1.3.0) | R Core Team | URL: https://tidyr.tidyverse.org/ |
| cowplot (v1.1.1) | CRAN | URL: https://cran.r-project.org/package=cowplot |
| readxl (v1.4.2) | CRAN | URL: https://cran.r-project.org/package=readxl |
| diffusr (v1.0.0) | CRAN | URL: https://cran.r-project.org/package=diffusr |
| ComplexHeatmap (v2.16.0) | Bioconductor | URL: https://bioconductor.org/packages/ComplexHeatmap |
| reshape2 (v1.4.4) | CRAN | URL: https://cran.r-project.org/package=reshape2 |
| grid (v4.3.2) | R Core Team | N/A (Part of base R) |
| ggraph (v2.1.0) | CRAN | URL: https://ggraph.data-impr.com/ |
| ggConvexHull (v0.1.0) | CRAN | URL: https://cran.r-project.org/package=ggConvexHull |
| patchwork (v1.1.2) | CRAN | URL: https://cran.r-project.org/package=patchwork |
| PCSF (v1.2.0) | Bioconductor | URL: https://bioconductor.org/packages/PCSF |
| readr (v2.1.4) | R Core Team | URL: https://readr.tidyverse.org/ |
| tidygraph (v0.3.0) | CRAN | URL: https://tidygraph.data-impr.com/ |
| fgsea (v1.26.0) | Bioconductor | URL: https://bioconductor.org/packages/fgsea |
| gprofiler2 (v0.1.1) | CRAN | URL: https://cran.r-project.org/package=gprofiler2 |
| limma (v3.56.2) | Bioconductor | URL: https://bioconductor.org/packages/limma |
| enrichR (v2.1) | CRAN | URL: https://cran.r-project.org/package=enrichR |
| data.table (v1.14.8) | CRAN | URL: https://cran.r-project.org/package=data.table |
| PhosR (v1.16.0) | Bioconductor | URL: https://bioconductor.org/packages/PhosR |
| pathview (v1.38.0) | Bioconductor | URL: https://bioconductor.org/packages/pathview |
| tidyverse (v2.0.0) | R Core Team | URL: https://www.tidyverse.org/ |
| edgeR (v3.42.0) | Bioconductor | URL: https://bioconductor.org/packages/edgeR |
| sva (v3.48.0) | Bioconductor | URL: https://bioconductor.org/packages/sva |

| Reagent/resource | Reference or source | Identifier or catalog number |
|---|---|---|
| psych (v2.3.6) | CRAN | URL: https://cran.r-project.org/package=psych |
| plotly (v4.11.0) | CRAN | URL: https://cran.r-project.org/package=plotly |
| scatterplot3d (v0.3-41) | CRAN | URL: https://cran.r-project.org/package=scatterplot3d |
| clusterProfiler (v4.8.0) | Bioconductor | URL: https://bioconductor.org/packages/clusterProfiler |
| enrichplot (v1.20.0) | Bioconductor | URL: https://bioconductor.org/packages/enrichplot |
| msigdbr (v7.5.1) | CRAN | URL: https://cran.r-project.org/package=msigdbr |
| RCy3 (v2.18.0) | Bioconductor | URL: https://bioconductor.org/packages/RCy3 |
| viridis (v0.6.3) | CRAN | URL: https://cran.r-project.org/package=viridis |
| ggvenn (v0.1.9) | CRAN | URL: https://cran.r-project.org/package=ggvenn |
| RColorBrewer (v1.1-3) | CRAN | URL: https://cran.r-project.org/package=RColorBrewer |
| tibble (v3.2.1) | R Core Team | URL: https://tibble.tidyverse.org/ |
| GGally (v2.1.2) | CRAN | URL: https://cran.r-project.org/package=GGally |
| decoupleR (v2.4.0) | Bioconductor | URL: https://bioconductor.org/packages/decoupleR |
| TCGAretriever (v1.16.0) | Bioconductor | URL: https://bioconductor.org/packages/TCGAretriever |
| plyr (1.8.6) | CRAN | URL: https://cran.r-project.org/package=plyr |
| dorothea (v1.6.0) | Bioconductor | URL: https://bioconductor.org/packages/dorothea |
| viper (v1.30.0) | Bioconductor | URL: https://bioconductor.org/packages/viper |
| progeny (v1.20.0) | Bioconductor | URL: https://bioconductor.org/packages/progeny |
| **Other** | | |
| RNeasy Mini Kit | Qiagen | Cat. No. 74104 |

| Reagent/resource | Reference or source | Identifier or catalog number |
|---|---|---|
| DNase I kit | Thermo Fisher | Cat. No. ENO525 |
| cDNA synthesis kit | Takara Bio | Cat. No. 6110A |
| NEBNext RNA Ultra II kit | NEB | Cat. No. E7770S |
| Phosphoplex assay | Protavio | N/A |
| PamChip® peptide microarrays | Pamgene International BV | N/A |

## Cell culture

Parental and *ARID1A*-KO A375 lines were grown in DMEM-F12 (Gibco, Cat. No.: 11320033) with 10% FBS (Gibco, Cat. No.: 10500-064) and 1% Penicillin–Streptomycin–Neomycin (Sigma, Cat. No.: P4083) at 37 °C with 5% $CO_2$. Cells were grown in a monolayer, and the culture medium was changed every 3 days. The cells were passaged once they reached ~80% confluence.

## Whole-genome CRISPR screen in the presence of trametinib

A genome-wide screen to identify genes conferring resistance to trametinib was performed using a human genome-wide library (Yusa V1), which targets ~18,000 genes with ~91,000 gRNAs using a detailed protocol described with reagents and product codes previously (Sharma and Wright, 2020). Two sets of Cas9-expressing A375 cells (80 million starting population for each set) were transduced at a MOI of 0.3 with the genome-wide lentiviral library. A day post infection, cells were treated with 2 µg/mL puromycin to remove any non-transduced cells. Live cells on day 7 were split into two populations; one was treated with 10 nM trametinib, and the control set was left untreated. All live cells post-treatment, and 30 million control untreated cells, were collected after 2 weeks, genomic DNA was extracted, gRNAs were amplified, and libraries were generated. The MAGeCK software (Li et al, 2014) was used to identify genes that were enriched in the live population compared to the control population.

## Luminex assay

For the Luminex assay, roughly 20,000 cells were seeded on 96-well plates overnight in 100 µL serum-starved medium (Cell culture media without FBS). The next morning, culture media containing serum (standard cell culture media) and supplemented with drugs or growth factors (as relevant) were added to the cells. After 6 h, the plate with cells was placed on ice, and the cells were washed with 100 µL of cold PBS. Next, 60 µL lysis mix was added to the cells, and the plate was shaken for 20 min at +4 °C and 650 rpm. The lysis mix was prepared by mixing Lysis buffer (Protavio, Cat. No.: PR-ASSB), Protease Inhibitors (1 tablet to 50 mL of Lysis buffer, Roche, Cat. No.: 11873580001), and 2 mM PMSF (Sigma, Cat. No: 93482). Cell lysates were then frozen at −20 °C and phosphorylation was measured utilizing the multiplex assay service provided by Protavio (Tsolakos et al, 2024) (Athens, Greece). We developed a phospho-plex platform

for semi-quantitative analysis of the phosphorylation status of 17 phosphoproteins (Table EV1), which displays a good signal-to-noise ratio to be measured in the in vitro assays (Table EV2).

The assay is based on the xMAP technology developed by Luminex Corporation. A mix of 17 capture antibodies coupled to Luminex magnetic beads (Bead mix) was prepared. Each antibody is coupled to a different magnetic bead region. Beads can be uniquely identified and differentiated by the Luminex instrumentation due to their unique color classification. A "detection" mix consisting of 17 biotinylated secondary antibodies specific for recognizing the analytes of the panel was also prepared. Detection antibodies are biotinylated, in order to be recognized by a streptavidin–phycoerythrin (SAPE) substrate used to produce the final detection signal.

Each sample is incubated with the bead mix in a well of a 96-well microtiter plate to allow binding of the analyte. Any unbound material is removed by washing using a magnetic separator. The formed antibody-analyte complex is incubated with the secondary detection antibody mix. Any unbound detection antibody is removed by a washing step, and the formed complex of capture antibody-analyte-detection antibody is labeled with SAPE (MOSS Inc., Pasadena, Maryland, USA, Cat. No: SAPE-001). The fluorescent emission of R-phycoerythrin and the distinct microsphere fluorescent signatures are measured simultaneously by the Luminex® instrument.

## Generation of ARID1A-KO cell line

ARID1A-KO cell lines were purchased from Synthego. Sequencing across the cut locus revealed a single nucleotide insertion, which led to a frame-shift mutation (Fig. EV1A). The KO efficiency as determined by TIDE was ~98% (Fig. EV1B) (Brinkman et al, 2014).

## Sample preparation for phosphoproteomics and functional kinomics analysis

Cells were seeded overnight at $0.5 \times 10^6$ cells/well density in six-well plates. Before harvesting, cells were treated with vemurafenib (APExBio, Cat. No.: A3004) (1 mM) and trametinib (APExBIO, Cat. No.: A3018) (10 nM), separately or in combination, for 6 h. Approximately $10^6$ cells were used as the starting material. Cells were washed twice with ice-cold PBS, scraped with a cell scraper, and then centrifuged at 1000×g for 3 min. The supernatant was removed, and the cell pellets were frozen in liquid nitrogen. For functional kinomics data generation, the same protocol was used except that $2 \times 10^6$ cells/T25 flasks were seeded overnight, and all cells were collected post-treatment.

## Kinome activity profiling through functional kinomics

Protein Tyrosine Kinase (PTK) and Serine/Threonine Kinase (STK) activity profiles were assessed using PamChip® peptide microarrays (Pamgene International BV, BJ 's-Hertogenbosch, The Netherlands), which measure the ability of active kinases in a protein lysate to phosphorylate specific peptides imprinted on multiple peptide arrays. A PTK PamChip® microarray contains 196 immobilized peptides, while a STK PamChip® microarray contains 140 peptides covalently attached to a porous material. Peptides harbor phosphorylation sites derived from literature or computational predictions that are associated with one or more upstream kinases. Phosphorylation is detected by (phospho-)specific primary antibodies, and the signal is quantified by

FITC-conjugated secondary antibodies. Detection is performed in multiple cycles at different exposure times and is monitored by a CCD camera. Protein lysates were prepared from untreated and drug-treated A375 melanoma cell lines of both genotypes, using M-PER mammalian extraction buffer (Thermo Fisher Scientific, Waltham, MA, USA), containing Phosphatase Inhibitor Cocktail and Halt Protease Inhibitor Cocktail EDTA-free (1:100 each; Thermo Fisher Scientific, Waltham, MA, USA), and quantified by the Bradford assay. A total of 5 µg and 1 µg of protein lysates per array from each biological sample were used to profile tyrosine and serine/threonine kinase activity, respectively, according to PamGene's standard protocols. Image analysis, peptide quality control (QC), signal quantitation, data normalization and visualization, as well as upstream kinase prediction were performed using Bionavigator v.6.3 software (PamGene, 's-Hertogenbosch, The Netherlands), according to manufacturer's instructions. Three independent biological replicates were used for each condition (parental (wild-type, WT), ARID1A-KO, untreated or drug-treated ARID1A-WT A375 melanoma cells).

## Transcriptomics data generation

Cells were seeded at $0.5 \times 10^6$ cells/well density in 6-well plates and left overnight. Before harvesting, cells were treated with Vemurafenib and Trametinib, separately or in combination, for 6 h. Approximately $10^6$ cells were used as the starting material. Cells were washed twice with ice-cold PBS, scrapped with a cell scraper, and then centrifuged at 1000×g for 3 min. For RNA isolation, the RNeasy Mini Kit (Qiagen, Cat. No. 74104) was used per manufacturer instructions, along with QIAshredder columns for the homogenization of cell lysates and DNase I treatment using DNase I kit (Thermo Fisher, Cat. No. ENO525). Following RNA extraction, the integrity of the isolated RNA was assessed using the Agilent 4200 TapeStation system (Agilent) according to the manufacturer's instructions, using 2 µL of a 1:150 dilution in $H_2O$ of each purified total RNA sample. Samples were tested further for cDNA synthesis using PrimeScript 1st strand cDNA synthesis kit (Takara Bio, Cat. No. 6110A) followed by RT-PCR on β-Actin using 2 µL of cDNA product. mRNA-seq libraries were prepared from 500 ng of total RNA with the NEBNext RNA Ultra II kit coupled with the poly(A) mRNA Magnetic Isolation Module on the Beckman liquid handling robot i7 with these parameters: 10 min fragmentation and 10 PCR cycles. Sequencing of all 36 samples in one pool with the Illumina sequencer NextSeq 2000 and the P3-50 kit.

## Transcriptomics data processing

We performed normalization of the raw counts using the voom function from the limma package (Ritchie et al, 2015). We selected only genes with a count-per-million (CPM) greater than 2 in at least one sample, to retain for downstream analysis. We computed scaling factors to convert observed library sizes into effective library sizes using the library edgeR (Robinson et al, 2010). We used these normalized counts as inputs to MOFA (Argelaguet et al, 2018). For visualization purposes, we used the package limma to predict stable results and to calculate significance provided for the presented volcano plots and LFCs (Fig. EV2A; Dataset EV4).

For transcription factor activity inference, we used the Python (version 3.8.18) package decoupleR. Differential expression was

computed using the Wald test, and log-fold changes (LFCs) were extracted using the pydeseq2 library. The CollecTRI gene regulatory network, focusing on human transcriptional regulation, was retrieved using the decoupler library. We employed the Univariate Linear Model (ULM) implemented in decoupler to infer transcription factor (TF) activities from differential expression data (Badia-I-Mompel et al, 2022).

## Proteomics/phosphoproteomics data generation

### Sample preparation

In total, 300 µL of a detergent-based buffer (1% sodium deoxycholate (SDC), 10 mM tris(2-carboxyethyl)phosphine (TCEP), 10 mM Tris, and 40 mM chloroacetamide) with cOmplete mini EDTA-free protease inhibitor cocktail (Roche Cat. No. 04693132001) was added to the cell lysates, boiled for 5 min at 95 °C, and sonicated using the Bioruptor for 20 cycles of 30 s on:30 s off. Protein quantification was carried out using the Bradford assay, and an aliquot corresponding to 200 µg was retained for each sample. In all, 50 mM ammonium bicarbonate was added, and digestion was allowed to proceed overnight at 37 °C using trypsin (Sigma T6567) and LysC (Wako, 125-05061) at 1:50 and 1:75 enzyme:substrate ratios, respectively. The digestion was quenched with 10% formic acid, and the resulting peptides were cleaned up in an automated fashion using the AssayMap Bravo platform (Agilent Technologies) with a correspond-ing AssayMap C18 reverse-phase column, followed by vacuum drying. To generate a reference channel to be used for all experiments, a pool of all samples combined was digested.

Dried peptides were re-solubilised in TMT resuspension buffer (87.5% HEPES, 12.5% ACN) and 11-plex TMT labels (Thermo Fisher A34808) were prepared according to the manufacturer's instructions. TMT labels were added to samples and labeling was allowed to occur during 2 h at RT (room temperature), after which the reaction was quenched using 5% hydroxylamine in HEPES, for 15 min at RT. The various channels were combined for each experiment, and the acetonitrile content was reduced by evapora-tion. The samples were then cleaned using a SepPak 1cc cartridge and dried completely before solubilising in HpH buffer A (10 mM NH$_4$OH, pH 10.8). HpH fractions were collected every minute over a 100 min gradient. Fractions between minutes 10 and 70 were concatenated into 20 fractions, and an aliquot of each was set aside for vacuum drying and full proteome analysis. The remainder of the concatenated fractions were vacuum-dried, followed by re-solubilization in 80% ACN/0.1% TFA for phosphopeptide enrich-ment. The enrichment was carried out in an automated fashion using the AssayMap Bravo platform (Agilent Technologies) with corresponding AssayMap Fe(III)-NTA cartridges, and eluates were dried by vacuum centrifugation and resolubilised in 1% FA, of which ~1 µg was injected on column.

### MS analysis

All spectra were acquired on an Orbitrap Exploris mass spectro-meter (Thermo Fisher Scientific BRE725535) coupled to an Ultimate 3000 liquid chromatography system. Peptides were trapped on a 300 µm i.d. × 5 mm C18 PepMap 100, 5 µm, 100 Å trap column (Thermo Scientific P/N 160434) and then separated on a 50 cm (75 µm ID) in-house packed column using Poroshell 120 EC-C18 2.7-Micron (ZORBAX Chromatographic Packing,

Agilent). Samples were eluted over a linear gradient ranging from 9 to 45% 80% ACN/0.1% FA over 65 min, 45 to 99% 80% ACN/0.1% FA for 3 min, followed by maintaining at 99% 80% ACN/0.1% FA for 5 min at a flow rate of 300 nL/min. Phosphopeptide-enriched samples were eluted over a linear gradient ranging from 9 to 34% 80% ACN/0.1% FA over 95 min, 45 to 99% 80% ACN/0.1% FA for 3 min, followed by maintaining at 99% 80% ACN/0.1% FA for 5 min at a flow rate of 300 nL/min. MS1 scans were carried out at a resolution of 60,000, with standard AGC target and automatic IT. The intensity threshold was set to 5.0e4, charge states 2–6 were included, and dynamic exclusion was used for a duration of 14 s. For the MS2 scans, a window of 1.2 $m/z$ was applied, with HCD collision energy set to 30%, an Orbitrap resolution of 45000, and standard AGC target and automatic maximum injection time determination.

### Raw data processing

All raw files were processed using ProteomeDiscoverer (Thermo) version 2.4.1.15, and the obtained data were searched against the SwissProt Homo Sapiens proteome (April 2021 release), using the Mascot proteomic search engine with the following settings: a maximum of two missed cleavages, precursor mass tolerance of 10 ppm, and fragment mass tolerance of 0.8 Da. Oxidation (M) and Phosphorylation (STY) were selected as dynamic modifications, with carbamidomethyl (C)- and TMT-tags (K and N-terminal) being selected as static modifications.

## Phosphoproteomics data processing

Data processing and analysis were conducted in R (version 4.2.0). We took raw peptide abundances and protein abundances from Proteome Explorer, and we normalized both by sample loading normalization. The normalized data were processed to extract phosphosite information for peptides, including amino acid residue and position. UniProt IDs and phosphosite information were parsed using the EnsDb.Hsapiens.v86 annotation R package (Rainer et al, 2019).

The data is sparse, so missing values were imputed across replicates within a single condition using the function scImpute from the R library PhosR (Kim et al, 2021). We performed this if there are three or more quantified values of that variable within a given condition (drug treatments and ARID1A-KO). Data were median-centered and scaled, and batch effects from the different TMT-11 plex runs were handled by the R package ComBat (Zhang et al, 2020). Only samples with quantified values above a threshold of 50 were retained for further analysis. Log2 transformations were applied to facilitate downstream statistical modeling.

To decouple the effects of total protein abundance on phosphorylation, a regression-based method was employed to estimate the "net" phosphorylation level (as implemented in (Roumeliotis et al, 2017)). This approach calculates residuals from a linear model in which log-transformed phosphoprotein abun-dance is regressed against the corresponding total protein abundance for each sample (*phosphopeptide abundance ~ protein abundance*). The resulting residuals represent phosphorylation changes independent of total protein levels. The processed data was aggregated based on unique phosphosite identifiers, with any duplicate entries being combined.

## Multi-omics factor analysis

To input the multi-omics data into Multi Omics Factor Analysis (MOFA) (Argelaguet et al, 2018), we used ANOVA to select for the most highly variable phosphopeptides and proteins. The criteria for phosphopeptides significance were an FDR-adjusted ANOVA $P$ value < 0.1 and an absolute mean abundance change >0.5. These were then standardized by z-score transformation. The criteria for protein and mRNA abundance were stricter (those with $P$ values < 0.001 and absolute values > 1). This helps in reducing noise by focusing on biologically relevant features.

A MOFA object was created using the integrated, long-format dataset, comprising phosphoproteomic, proteomic, and mRNA expression views. Default data, model, and training options were configured using the R package MOFA2 (Argelaguet et al, 2020). The number of latent factors was set to 3, reflecting the number of sources of variation in the data (Figs. 1 and EV3). After training, sample metadata (e.g., drug treatment and genetic background) were integrated into the MOFA object, enabling stratified analysis based on biological conditions. The feature loadings for Factor 1 were multiplied by -1, such that in all factors negative loadings represent features that correspond to control samples (untreated, non-combination therapy or WT in factors 1, 2, and 3, respectively).

## Incucyte-based viability analysis

gRNAs targeting empty or EGFR were lentivirally transduced to mOrange and Cas9 expressing parental and KO cell lines. After 7 days, cells were seeded in 96-well plates at 40,000 cells/well and allowed to adhere overnight. Drugs were added the next day in 100 μL medium, and plates were placed in an Incucyte for 96 h, with images acquired every 4 h to monitor mOrange fluorescence as a measure of cell viability.

Total mOrange intensity per well was extracted and smoothed using a rolling median (window = 3) with MAD-based spike removal and linear interpolation. Each well was baseline-normalized by subtracting its initial ($t_0$) value, and mean ± SEM trajectories were calculated across replicates. Δ(Drug − DMSO) values were computed at each timepoint, and final (96 h) differences between ARID1A-WT and ARID1A-KO lines were assessed using Welch's $t$ test.

## Network propagation using phuEGO

To explore the functional interactions between proteins identified from the MOFA factors, we employed phuEGO, a tool designed for signaling network construction (Giudice et al, 2024). The goal was to identify potential regulatory hubs and pathways that are influenced by the key proteins associated with the latent factors across all omics views in our dataset. Latent factor weights were extracted for each omics view (i.e., phosphoproteomics, proteomics, and mRNA expression) using the get-weights function from the MOFA2 package. For each protein, the phosphosite with the highest absolute weight was selected, and then the results of each condition were aggregated across each view, so we had a protein-level weight. We extracted proteins exhibiting the most extreme weights for each factor by selecting the top or bottom 5% quantiles. Using these proteins as seeds, we ran phuEGO with parameters

Fisher-threshold = 0.1, Fisher-background = intact, Random-walk damping = 0.95, RWR-threshold = 0.01, and KDE-cutoff = 0.5. This generated a network for each factor found in the multi-omics data. PhuEGO accounts for hub biases in the network propagation by comparing results to 1000 perturbed networks with preserved node degree distribution. Further Laplacian normalization of the adjacency matrix dampens the influence of highly connected hubs. These steps ensure that network results are not the result of hub artefacts.

## Prize-collecting Steiner–Forest analyses

To identify key functional genes within the larger factor networks, we employed the Prize-Collecting Steiner–Forest (PCSF) algorithm with randomized edge costs. The randomized method enhances the robustness of the resulting sub-network by running the algorithm multiple times, while adding random noise to the edge costs (Akhmedov et al, 2016). We selected the top 50 central genes as terminal nodes based on their PageRank centrality, which has been proven to produce biologically meaningful results (Brin and Page, 1998; Iván and Grolmusz, 2011; Barker et al, 2022; Garrido-Rodriguez et al, 2024). These nodes were then weighted as prizes, and then the PCSF algorithm was run with 4000 iterations, with up to 5% random noise added to edge costs.

## Random walk of receptors upregulated in ARID1A-KO cancer cells

We implemented an algorithm for analyzing receptor-mediated pathways using network propagation (via random walks), with a focus on perturbations caused by ARID1A-KO and combination drug treatments in ARID1A-KO conditions. We used the union of graphs for Factor 1 (showing drug agnostic responses) and Factor 3 (showing ARID1A KO-induced responses) to visualize the inter-action between these two processes. For both ARID1A KO and combination drug treatments, we extract receptor LFC (log-fold change) data and split them into upregulated and downregulated receptors. We processed LFC datasets from differential expression analysis by filtering receptors (as defined by Omnipath) that are significantly affected (adjusted $P$ value < 0.01). Receptors were identified using the OmnipathR package. These LFCs are normal-ized by dividing each element by the sum of the vector (to prepare for random walks). We then perform a random walk on a graph with a given starting vector of probabilities derived from the LFCs above. The random walk is corrected for hubs and used to compute a stationary distribution that indicates the relative importance of a node for each condition.

To study the effect of increases in Ephrin receptor activation on our network, we take the union network prior to PageRank/PCSF pre-processing and perform a random walk with restart on the receptor kinase activity, determined by the functional kinomics, target-peptide, phospho-tyrosine data. We identified receptor proteins in the kinomics phospho-tyrosine data using the OmnipathR package. We filtered this data to retain kinases that had statistically significant changes in activity (e.g., higher than a threshold of ±1 median kinase activity). Starting from an initial set of deregulated receptors (normalized from the previous step), we performed random walks over the network to estimate the stationary distribution of each node. To assess the significance of

observed distributions, we conducted permutation tests by generating 10,000 random networks that preserved the degree distribution of the original network. We compared the original and randomized distributions to calculate $P$ values.

## Maximum flow of receptors to nodes of interest

We selected the nodes that were the most prominent in the heat diffusion from both combination therapy and ARID1A KO (cytosolic proteins, JUN, PRKD1, PTPN6, and FYN) and performed heat diffusion individually from each upregulated receptor in both conditions to detect which receptors were responsible for their flagging (Fig. EV5A). To identify pathways in the network that are responsible for the highlighted network propagations between these receptors (ROS1, FGFR1, and EGFR) and their targets, we computed the maximum flow between a specified source and sink node from the combined graph, where edge capacities were defined by semantic similarity. This is to preferentially flow the signal through nodes that are more functionally similar. This was done using the R package, igraph (Csardi and Nepusz, 2006).

We extracted the flow values from the maximum flow result and identified edges with flow values exceeding the 95th percentile. For visualization, a sub-graph was induced from the original graph, retaining only the nodes connected by high-flow edges. We identified the connected components in the sub-graph and selected the component containing both the source and sink nodes. This sub-graph represents the largest connected high-flow region within the network between modulated receptors and nodes of interest.

## Clinical analysis of *ARID1A* gene-perturbed patients

Melanoma RNA-seq data and corresponding mutation information were retrieved from TCGA (The Cancer Genome Atlas) (The Cancer Genome Atlas Research Network, 2013) using the TCGA retriever package. We extracted mRNA expression data (FPKM values) for melanoma patients using the fetch_all_tcgadata() function. We obtained *ARID1A* mutation and CNV (copy number variation) status to identify samples with *ARID1A* alterations. We then categorized samples as having *ARID1A* mutations, deletions, or neither (wild-type, WT) (Appendix Fig. S5). To determine transcription factor activity, we employed the decoupleR package, using the Univariate Linear Model (ULM) as described above.

We also performed differential expression analysis to identify dysregulated genes in *ARID1A*-mutant versus ARID1A-WT melanoma patients using the limma package. We filtered out low-expression genes (minimum count = 10) using the filterByExpr using the edgeR package. We log-transformed the data and normalized them via engagement of the voom function to stabilize the variance. We then fitted a linear model to identify the differentially expressed genes and identified significant genes on the adjusted $P$ values (FDR < 0.01).

## Enrichment analyses

EnrichR (available https://maayanlab.cloud/Enrichr/) was used to perform enrichment analysis of the members of the network using the BioPlanet 2019 database.

## Data visualization

All analyses were performed in R or Python, with the code being available upon request. The MOFA analysis was conducted using the MOFA2 package, with additional processing and visualization supported by dplyr, purrr, stringr, ggplot2, PhosR, and EnsDb.Hsapiens.v86.

The ggraph package was used to visualize the receptor networks and the results of the network propagation analysis (Pedersen).

## Data availability

The code required for the analysis is available on Github (https://github.com/CharlieBarker/Melanoma_Resistance). Proteomics and phosphoproteomics data has been deposited in PRIDE under the identifier PDX058857. Gene expression data has been deposited in Expression Atlas through Biostudies and Annotare under the identifier E-MTAB-14759.

The source data of this paper are collected in the following database record: biostudies:S-SCDT-10_1038-S44320-025-00183-5.

## Peer review information

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

## Acknowledgements

The work was supported by the European Molecular Biology Laboratory (CGB, SS, EP), the EMBL International PhD Programme (CGB), and Wellcome Trust (SS, CG-T; grant number 317519/Z/24/Z to SS and 206194 to GJW); The EMBL GeneCore Facility is acknowledged for support in transcriptomics data collection. The results in Fig. 5D are based upon data generated by the TCGA Research Network: https://www.cancer.gov/tcga. EMBL IT Support is acknowledged for the provision of computer and data storage servers. The proteomics and phosphoproteomics data collection was supported by the EU Horizon 2020 program INFRAIA project Epic-XS (Project 823839).

## Author contributions

**Charlie George Barker**: Conceptualization; Software; Formal analysis; Investigation; Visualization; Methodology; Writing—original draft; Writing—review and editing. **Sumana Sharma**: Conceptualization; Formal analysis; Funding acquisition; Validation; Investigation; Visualization; Methodology; Writing—original draft; Writing—review and editing. **Ana Mafalda Santos**: Validation; Investigation. **Konstantinos-Stylianos Nikolakopoulos**: Investigation. **Athanassios D Velentzas**: Formal analysis; Investigation. **Cristina Tormo-Garcia**: Validation. **Anushka Sharma**: Validation. **Franziska I Völlmy**: Investigation. **Angeliki Minia**: Validation. **Vicky Pliaka**: Validation. **Joseph Clarke**: Resources. **Maarten Altelaar**: Resources; Supervision. **Gavin J Wright**: Resources. **Leonidas G Alexopoulos**: Resources; Validation. **Dimitrios J Stravopodis**: Resources; Supervision; Writing—review and editing. **Evangelia Petsalaki**: Conceptualization; Resources; Supervision; Funding acquisition; Visualization; Methodology; Writing—original draft; Project administration; Writing—review and editing.

Source data underlying figure panels in this paper may have individual authorship assigned. Where available, figure panel/source data authorship is listed in the following database record: biostudies:S-SCDT-10_1038-S44320-025-00183-5.

## Funding

## Disclosure and competing interests statement

LGA is a co-founder of Protavio LTD. The authors declare no competing interests.

# Expanded View Figures

**Figure EV1.  ARID1A was effectively knocked out and affects signaling dynamics.**

(**A**) ARID1A and matched parental melanoma A375 cells were purchased from Synthego. Sanger sequencing traces for the edited region is shown. The single base-pair deletion led to a frame-shift mutation in the protein. (**B**) Output from TIDE analysis showing mismatches in the inference window (left panel) and 98% editing efficiency. (**C**) Killing of parental and ARID1A cell lines in the presence of Vermurafenib, Trametinib, combination of both drugs, and control with DMSO. Starting concentration of 30 nM Vermurafenib and 3 nM Trametinib was used (Left panel). The numbers on x-axis indicate the dilution factor (1:3) from the starting concentration. Images of cells at 90 h post treatment with the drugs (Right panel). (**D**) Luminex-based phosphoprotein measurements of indicated phosphosites on A375 cells treated either with Trametinib (10 nM) for indicated times, or with growth factors (TGFb and EGF) for 4 h. Long term indicates treatment with drug at 1 nM for 2 weeks. Log-fold change is calculated from unstimulated/ untreated condition. Decrease in MEK phosphorylation post trametinib treatment is observed between 1 and 6 h.

▶

 

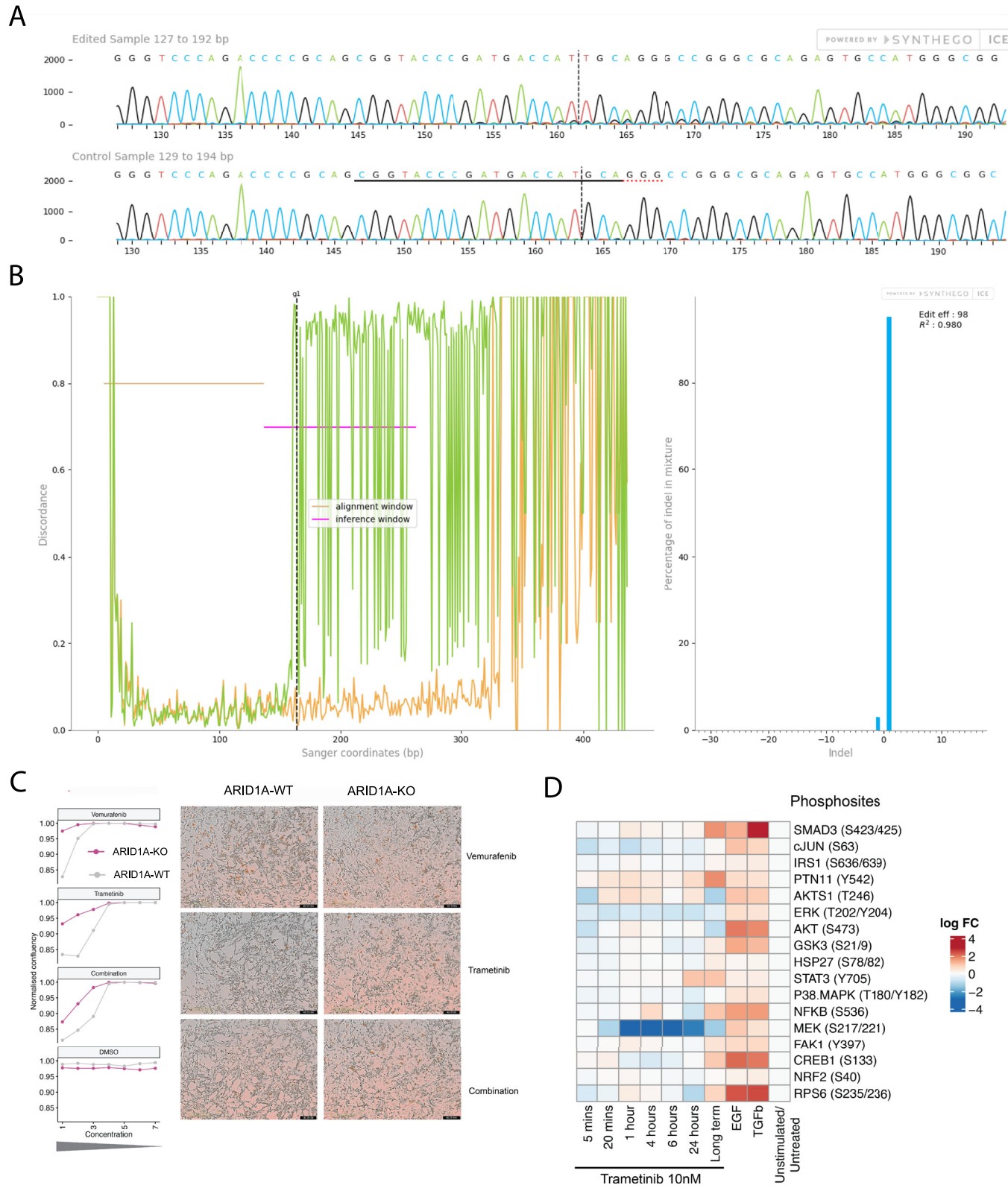

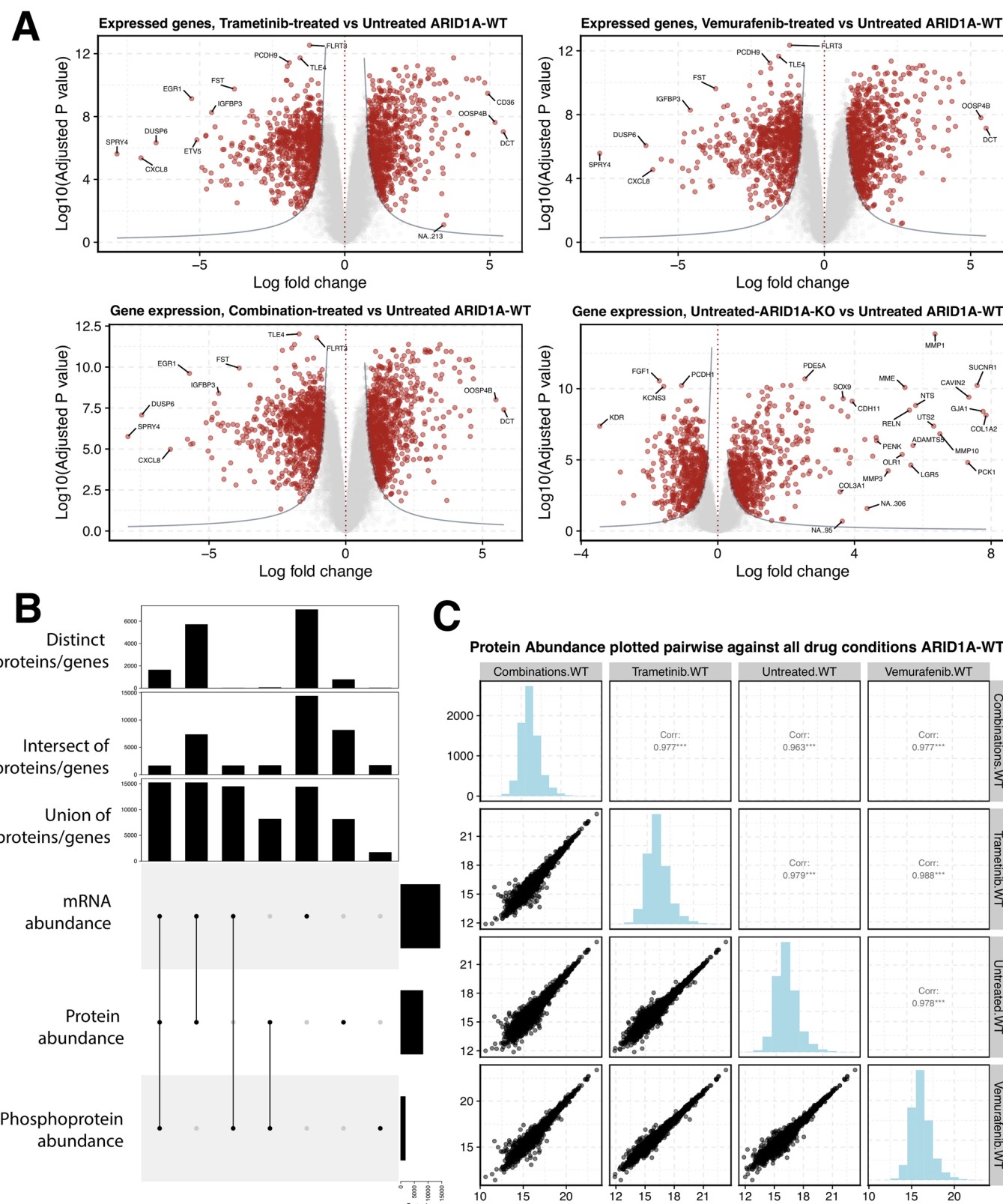

**Figure EV2.   Differential expression analysis and quantified proteins/genes across different 'omics data.**

(**A**) Volcano plots for abundant gene transcripts showing significance (FDR-adjusted *P* value, *y* axis) and log-fold change (*x* axis). LFCs for this data as well as phosphopeptides and proteins are available in the EV datasets. (**B**) Upset plot showing the extent of distinct proteins, genes and peptides quantified from each 'omics experiment. Barplots show the number of readings counted in each intersection of the data. (**C**) Plots showing the relation between protein abundance for isolated drug treatments.

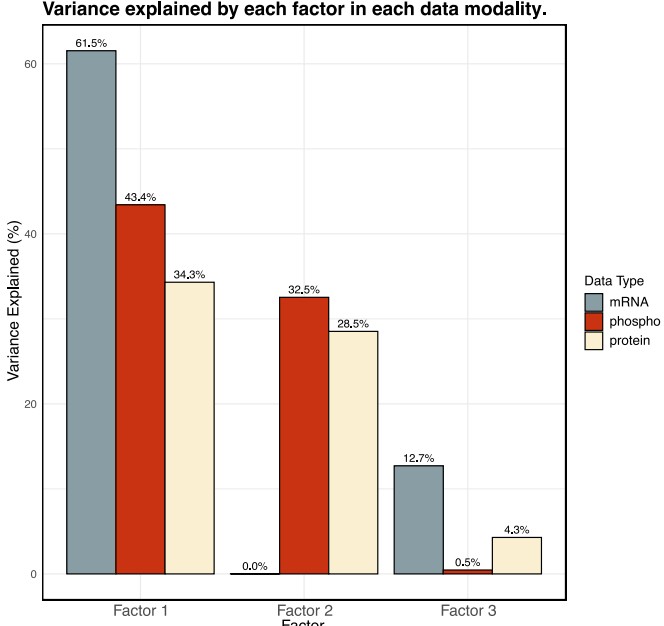

**Figure EV3.  Variance decomposition derived from MOFA analysis.**

Barplot showing the variance decomposition, assessing the proportion of variance explained by each factor in each 'omics modality.

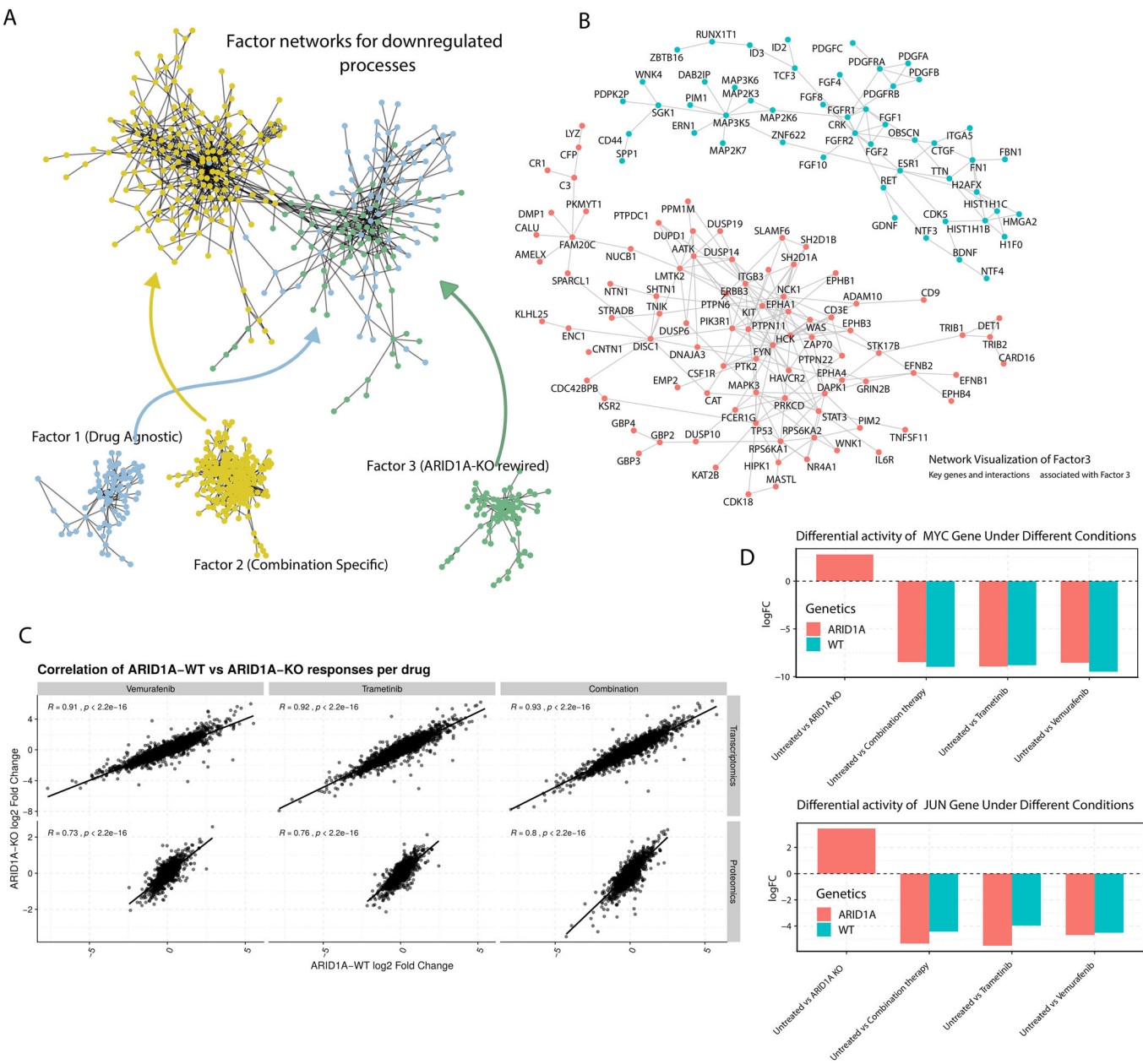

Figure EV4. **ARID1A-rewired network-based changes.**

(**A**) Graph showing the increased connectivity of nodes from both Factor 1 (blue) and Factor 3 (green) compared to Factor 2 (yellow). (**B**) PhuEGO-derived Network underlying changes associated with Factor3. Nodes represent proteins and the edges between them represent known interactions between those proteins. Red indicates networks associated with upregulated processes and blue indicates networks associated with downregulated processes. (**C**) Scatter plots showing the correlation between ARID1A-KO (*y* axis) and ARID1A-WT (*x* axis) drug response $\log_2$ fold changes in transcriptomic (top row) and proteomic (bottom row) data across different drug conditions (columns). Pearson correlation coefficients (r) and significance values (p-values) were calculated using the Pearson product–moment correlation test, in which significance is assessed using a t-distribution test of the null hypothesis that the true correlation is zero. (**D**) Plot showing the change in activation of TFs MYC (top) and JUN (bottom), based on the relative abundance of their known regulons as calculated using the ColectTRI database. This shows the difference in condition between ARID1A KO A375 cell lines (red) and parental cell lines (blue).

## A

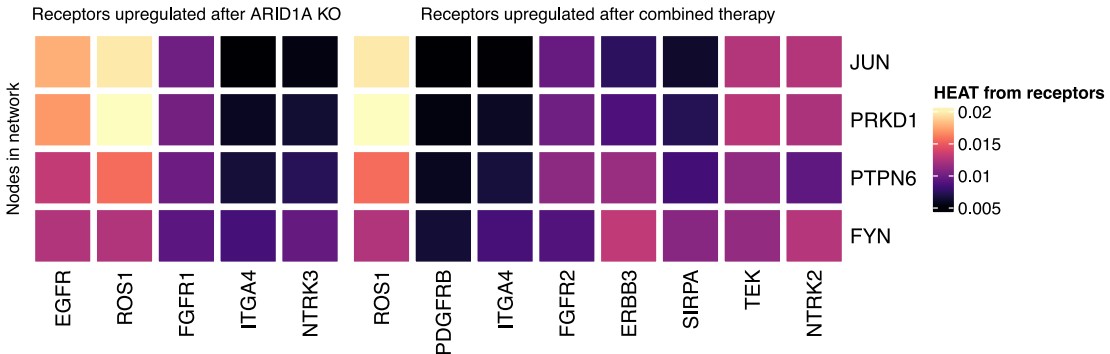

## B

**Flow Through Nodes in Signaling Paths from EGFR, ROS1 to JUN**

**Figure EV5. Network propagation illustrating the effect of dysregulated receptors.**

(A) Heatmap showing network propagation from individual receptors (columns) and their propagation to selected nodes (rows). (B) Barplot showing the flow (x axis) through nodes (y axis) within the combined Factor1/3 network from EGFR and ROS1 to JUN, ranked in order of flow.

