## [Peer Review File · Molecular Systems Biology]

Integrative multi-omics defines melanoma drug response and ARID1A-dependent resistance mechanisms

Charlie Barker, Sumana Sharma, Ana Mafalda Santos, Konstantinos-Stylios Nikolakopoulos, Athanassios Velentzas, Cristina Tormo-Garcia, Anushka Sharma, Franziska Völlmy, Angeliki Minia, Vicky Pliaka, Joseph Clarke, Maarten Altelaar, Gavin Wright, Leonidas Alexopoulos, Dimitrios J. Stravopodis, and Evangelia Petsalaki

Corresponding author(s): Evangelia Petsalaki (petsalaki@ebi.ac.uk)

Review Timeline:

Submission Date:	25th Mar 25
Editorial Decision:	18th May 25
Revision Received:	7th Nov 25
Editorial Decision:	5th Dec 25
Revision Received:	9th Dec 25
Accepted:	10th Dec 25

Editor: Jingyi Hou

Transaction Report:

18th May 2025

Manuscript Number: MSB-2025-12988

Title: ARID1A-induced transcriptional reprogramming rewires signalling responses to melanoma drug treatment

Author: Charlie Barker

Sumana Sharma

Ana Mafalda Santos

Konstantinos-Stylianos Nikolakopoulos

Athanassios Velentzas

Franziska Völlmy

Angeliki Minia

Vicky Pliaka

Maarten Altelaar

Gavin Wright

Leonidas Alexopoulos

Dimitrios J. Stravopodis

Evangelia Petsalaki

Dear Evangelia,

Thank you for submitting your work to Molecular Systems Biology. I apologize for the somewhat slow process, which was due to delays in obtaining the referee reports. We have now heard back from the three reviewers who agreed to evaluate your manuscript. As you will see from the reports below, the reviewers think the study potentially interesting. However, they raised a series of concerns, which we would ask you to address in a major revision.

I think the reviewers' recommendations are clear, so there is no need to reiterate the points listed below. Notably, all three reviewers emphasize the need for additional in vitro experimental validation to reinforce the study's conclusions and enhance the biological insights. This should be addressed carefully. Additionally, in response to Reviewer #2's comment, please more clearly emphasize the methodological novelty and advancement, particularly highlighting its advantages over standard data analysis workflows in uncovering novel biological insights.

All other issues raised by the reviewers need to be satisfactorily addressed as well. As you may already know, our editorial policy allows in principle a single round of major revision, so it is essential to provide responses to the reviewers' comments that are as complete as possible. Please feel free to contact me in case you would like to discuss in further detail any of the issues raised by the reviewers.

On a more editorial level, we would ask you to address the following issues:

- Please provide a .docx formatted version of the manuscript text (including legends for main figures, EV figures and tables). Please make sure that the changes are highlighted to be clearly visible.
- Please provide individual production quality figure files as .eps, .tif, .jpg (one file per figure).
- Please provide a .docx formatted letter INCLUDING the reviewers' reports and your detailed point-by-point responses to their comments. As part of the EMBO Press transparent editorial process, the point-by-point response is part of the Review Process File (RPF), which will be published alongside your paper.
- Please note that all corresponding authors are required to supply an ORCID ID for their name upon submission of a revised manuscript.
- We replaced Supplementary Information with Expanded View (EV) Figures and Tables that are collapsible/expandable online (see examples in <http://msb.embopress.org/content/11/6/812>). A maximum of 5 EV Figures can be typeset. EV Figures should be cited as 'Figure EV1, Figure EV2' etc... in the text and their respective legends should be included in the main text after the legends of regular figures.

Additional Tables/Datasets should be labeled and referred to as Table EV1, Dataset EV1, etc. Legends have to be provided in a separate tab in case of .xls files. Alternatively, the legend can be supplied as a separate text file (README) and zipped together with the Table/Dataset file.

For the figures and tables that you do NOT wish to display as Expanded View figures, they should be bundled together with their legends in a single PDF file called *Appendix*, which should start with a short Table of Content. Each legend should be below

the corresponding Figure/Table in the Appendix. Appendix figures and tables should be referred to in the main text as: "Appendix Figure S1, Appendix Figure S2, Appendix Table S1" etc. See detailed instructions regarding expanded view here: <https://www.embopress.org/page/journal/17444292/authorguide#expandedview>.

-Before submitting your revision, primary datasets (and computer code, where appropriate) produced in this study need to be deposited in an appropriate public database (see [http://msb.embopress.org/authorguide - dataavailability](http://msb.embopress.org/authorguide-dataavailability) <https://www.embopress.org/page/journal/17444292/authorguide#dataavailability>).

The accession numbers and database should be listed in a formal "Data Availability" section (placed after Materials & Method) that follows the model below (see also <https://www.embopress.org/page/journal/17444292/authorguide#dataavailability>). Please note that the Data Availability Section is restricted to new primary data that are part of this study.

Data availability

- RNA-Seq data: Gene Expression Omnibus GSE46843 (<https://www.ncbi.nlm.nih.gov/geo/query/acc.cgi?acc=GSE46843>)

- [data type]: [name of the resource] [accession number/identifier/doi] ([URL or identifiers.org/DATABASE:ACCESSION])

-At EMBO Press we ask authors to provide source data for the main figures. Our source data coordinator will contact you to discuss which figure panels we would need source data for and will also provide you with helpful tips on how to upload and organize the files.

- Our journal encourages inclusion of *data citations in the reference list* to directly cite datasets that were re-used and obtained from public databases. Data citations in the article text are distinct from normal bibliographical citations and should directly link to the database records from which the data can be accessed. In the main text, data citations are formatted as follows: "Data ref: Smith et al, 2001". In the Reference list, data citations must be labeled with "[DATASET]". A data reference must provide the database name, accession number/identifiers and a resolvable link to the landing page from which the data can be accessed at the end of the reference. Further instructions are available at .

- We updated our journal's competing interests policy in January 2022 and request authors to consider both actual and perceived competing interests. Please review the policy <https://www.embopress.org/competing-interests> and update your competing interests if necessary.

Please use the heading "Disclosure statement and competing interests".

- All Materials and Methods need to be described in the main text using our 'Structured Methods' format. According to this format, the Methods section includes a Reagents and Tools Table (listing key reagents, experimental models, software and relevant equipment and including their sources and relevant identifiers) followed by a Methods and Protocols section describing the methods, ideally using a step-by-step protocol format. The aim is to facilitate adoption of the methodologies across labs.

Please download and fill our Reagents and Tools Table template (.docx), which you can find in our author guidelines:

<https://www.embopress.org/page/journal/17444292/authorguide#structuredmethods>.

-Regarding data quantification:

Please ensure to specify the name of the statistical test used to generate error bars and P values, the number (n) of independent experiments (please specify technical or biological replicates) underlying each data point and the test used to calculate p-values in each figure legend. Discussion of statistical methodology can be reported in the materials and methods section, but figure legends should contain a basic description of n, P and the test applied.

Graphs must include a description of the bars and the error bars (s.d., s.e.m.).

- Please provide a "standfirst text" summarizing the study in one or two sentences (approximately 250 characters, including space), three to four "bullet points" highlighting the main findings and a "synopsis image" (550px width and 400-600 px height, PNG format) to highlight the paper on our homepage.

Here are a couple of examples:

<https://www.embopress.org/doi/10.15252/msb.20199356>

<https://www.embopress.org/doi/10.15252/msb.20209475>

<https://www.embopress.org/doi/10.15252/msb.209495>

When you resubmit your manuscript, please download our CHECKLIST (<https://www.embopress.org/pb-assets/embo->

site/EMBO%20Press%20Author%20Checklist-1642513524327.xlsx) and include the completed form in your submission. *Please note* that the Author Checklist will be published alongside the paper as part of the transparent process (<https://www.embopress.org/page/journal/17444292/authorguide#transparentprocess>).

If you feel you can satisfactorily deal with these points and those listed by the referees, you may wish to submit a revised version of your manuscript. Please attach a covering letter giving details of the way in which you have handled each of the points raised by the referees. A revised manuscript will be once again subject to review and you probably understand that we can give you no guarantee at this stage that the eventual outcome will be favorable.

I look forward to receiving the revised manuscript soon.

Kind regards,
Jingyi

Jingyi Hou, PhD
Senior Editor
Molecular Systems Biology

We realize that it is difficult to revise to a specific deadline. In the interest of protecting the conceptual advance provided by the work, we recommend a revision within 3 months (16th Aug 2025). Please discuss the revision progress ahead of this time with the editor if you require more time to complete the revisions. Use the link below to submit your revision:

IMPORTANT: When you send your revision, we will require the following items:

1. the manuscript text in LaTeX, RTF or MS Word format
2. a letter with a detailed description of the changes made in response to the referees. Please specify clearly the exact places in the text (pages and paragraphs) where each change has been made in response to each specific comment given
3. three to four 'bullet points' highlighting the main findings of your study
4. a short 'blurb' text summarizing in two sentences the study (max. 250 characters)
5. a 'thumbnail image' (550px width and max 400px height, Illustrator, PowerPoint or jpeg format), which can be used as 'visual title' for the synopsis section of your paper.
6. Please include an author contributions statement after the Acknowledgements section (see <https://www.embopress.org/page/journal/17444292/authorguide>)
7. Please complete the CHECKLIST available at (<https://bit.ly/EMBOPressAuthorChecklist>). Please note that the Author Checklist will be published alongside the paper as part of the transparent process (<https://www.embopress.org/page/journal/17444292/authorguide#transparentprocess>).
8. When assembling figures, please refer to our figure preparation guideline in order to ensure proper formatting and readability in print as well as on screen:
<https://bit.ly/EMBOPressFigurePreparationGuideline>
See also figure legend guidelines: <https://www.embopress.org/page/journal/17444292/authorguide#figureformat>
9. Please note that corresponding authors are required to supply an ORCID ID for their name upon submission of a revised manuscript (EMBO Press signed a joint statement to encourage ORCID adoption). (<https://www.embopress.org/page/journal/17444292/authorguide#editorialprocess>)
Currently, our records indicate that the ORCID for your account is 0000-0002-8294-2995.

Please click the link below to modify this ORCID:
Link Not Available

11. Include a Reagents and Tools Table, which can be downloaded from our author guidelines (<https://www.embopress.org/page/journal/17444292/authorguide#structuredmethods>)

*** PLEASE NOTE *** As part of the EMBO Press transparent editorial process initiative (see our Editorial at <https://dx.doi.org/10.1038/msb.2010.72>), Molecular Systems Biology publishes online a Review Process File with each accepted manuscripts. This file will be published in conjunction with your paper and will include the anonymous referee reports, your point-by-point response and all pertinent correspondence relating to the manuscript. If you do NOT want this File to be published, please inform the editorial office at contact@molsystbiol.org within 14 days upon receipt of the present letter.

Reviewer #1:

In this work, Barker et attempt to infer mechanisms of TKI resistance (Trametinib, Vemurafenib and combination of the two) in sensitive melanoma A375 and in resistant ARID1A-KO A375 cell lines. They performed multi-modal characterisation (transcriptomics, proteomics and phosphoproteomics) and performed MOFA analysis to identify three key emerging factors: Factors 1 and 2 capture adaptive responses to drug treatments whereas Factor 3 reflects sustained resistance due to ARID1A loss. On top of this performed complex network analysis using the Phuego Tool, developed in the group, to identify topological network structures involved in individual processes, for each key factor they provided biological conclusions. In particular, authors hypothesize in resistant ARID1A-KO A375 cell lines treated with TKI a reactivation of the MAPK pathway by the EGFR and ROS1 receptors converging on JUN via FYN and PRKD1. Also, they propose a role of ARID1A-mediated resistance in suppressing immunogenicity of melanoma cancer cells, with clear critical implication for immunotherapy in this type of cancer. The central conclusions are generally supported by a good and body of analyses and the study contributes to shedding light on resistance mechanisms in case of ARID1A loss (which constitute almost 10% of melanoma cases). For these reasons, I sincerely believe that the study will appeal to systems biologists and I support the publication.

However, I have the impression that the manuscript presentation lacks clarity and require a revision of text and figures: the many types of comparisons (ARID1A-KO vs parental, Vemurafenib-ctrl, Trametinib -ctrl, Combination-ctrl, treated ARID1A-KO vs treated parental., etc...) really hampers the readability and the clarity of the manuscript, in some instances, the interpretation could be better contextualized or clarified, the presentation of comparisons throughout the figures is at times unclear or incomplete, which hinders interpretability. Some conclusions-such as the suppression of immunogenicity-would benefit from more direct supporting evidence or elaboration and perhaps in vitro validation. Finally, the translational significance could be further highlighted, especially in terms of clinical relevance or therapeutic implications.

MAJOR REVISIONS:

- The title of the work is ARID1A-induced transcriptional reprogramming rewires signalling responses to drug treatment in melanoma, however we end up in this only in Figure 4, despite I believe that also the first part is interesting, authors should clarify what is the relevance to the present manuscript of the first paragraphs.
- The paragraph "The basal transcriptional state of cells following ARID1A KO influences response to drug treatment" looks to me confusing and I struggled in following text statements in the figures. Also, I am missing the point... If this cells are strongly resistant, why should they display a signalling rewiring after treatment? Also, did the author check whether drugs effectively are capable to enter ARID1A KO cells? This can in principle explain why there is no rewiring of the signalling.

LACK OF CLARITY: There are many comparison groups, it's difficult to follow which comparison is displayed/discussed, I strongly encourage the authors to make an extra effort in text and especially in Figures, where often this info is missing.

- I suggest the author to consistently refer in text and figure to ARID1A KO cells, as ARID1A-KO (some time it's just ARID1A (see as an example EV1) which is confusing).
- It's difficult for a non-expert of some of the analyses and tools to follow some parts. For example, it's unclear to me in Fig. 1C (Phuego network) what up/down mean...
- Similarly, in Network in Figure EV5, what are the up- or down-regulated processes? Is a Phuego output? It's the proteins' FoldChange? In case Fc in which condition or dataset?
- Overall, the work jump from drug-control to ARID1A-KO - parental analyses, which makes the readability difficult, can the work be divided in clear distinct paragraph?
- Lines 276-278, about the statement "their transcriptomic and proteomic drug response networks remain similar to parental cells (Fig. 1D; Fig. EV7A, EV7B)", I cannot find, in mentioned figures, the reported information. Which comparisons are you considering? ARID1A-KO vs parental or ARID1A-KO(treated) vs parental(treated)?

OTHER

- At the phosphoproteomic level are direct substrate of BRAF or MEK changing? When checking for this I noticed that the phosphoproteomic dataset ST3 has only 900 rows, where only one displays a significant adj-pvalue, in text I read that there should be more, please check the table is correct.
- In Table S2-4, Authors should add also statistics for all the comparisons included in the study.
- Fig. 2C is rich of information, however is not discussed in text. About Fig.2C, FC in abundance to which comparison refers to? All treatment vs untreated? Please add info... (Similarly, add info to and Fig. 3B.) Also, what does 'semantic similarity' indicate? Is it relevant?
- Line 231, about "Despite phosphoproteomics being the main driver for this factor", where did you get this info from?

- Lines 233-236 "Looking at the phosphosites that drive the variance captured by Factor 2, we find several phosphoproteins involved in DNA repair-related functions (Fig. 3C and Table S5)." How did the authors find this information? Can they quantify? which are these DNA-repair proteins (only the ones in Fig. 3C)? Are there other processes? .
- Lines 298-300, authors should exclude that there is HUBS bias here, I suggest to do a randomisation.
- The model in Figure 4F is very nice, could it be integrated with literature info, to add directionality to the edges, since we are describing a signalling network. I think this could really add information. To conclude, have authors considered to perform in vitro validation of this results?

MINOR POINTS:

- Referring to lines 132-135, I have the impression that EV3B refers to the entire dataset and not to the significant ones.
- About the "correlates strongly" in line 189, I acknowledge that the R tend to be good, however, I would reduce the statement as it's significant in only one case (EV5D) .
- Line 230, please replace MAPK2K1 with MEK, as you used this nomenclature in the rest of the manuscript.

Reviewer #2:

Summary

In this manuscript, Barker et al used graph methodology to investigate mechanisms of resistance to MAPK pathway inhibitors in a ARID1A-KO melanoma cell line relative to its WT counterpart. To this end, the authors integrate transcriptomic, proteomic and phosphoproteomics data using innovative approaches, some of which were developed by the Petsalaki group. The authors used multiomic factor analysis to identify three "factors" that drive variation across the RNA-seq, proteomics and phosphoproteomics data. The first two factors capture adaptive response to treatments while Factor 3 capture resistance due to ARID1A loss. The loadings from MOFA were then used as input of a network propagation method named phuEGO, which was recently developed by the group. The authors conclude the elevated RTKs such as EGFR and ROS1 and the downstream signaling nodes PRKD1, JUN and NCK1 contribute to resistance.

General remarks

The study seems to be well conducted from the computational standpoint and the authors place emphasis on the use of innovative approaches to multiomic data analysis. It is not clear, however, whether such new methods (phuEGO, MOFA) add insights that could not have been obtained with the use of standard means to analyze omics data. I think showing this is of key importance for the paper to have novelty from the methodological standpoint. On the other hand, if the authors claim that the study is not a methods paper, and instead it shows new biological insights, then more work should be done to validate such new knowledge. In summary, the impression of this reviewer, from the way that the manuscript is presented, is that the study falls short of having methodological or validated biological novelty.

Major points

1. As pointed above, the authors should show that complex graph-theoretical methodology offers advantages for the analysis of omics data. What information has this analysis provided that could not have been obtained from a standard analysis of the data?
2. The major issue, aside from the point above, is that the authors propose the involvement of several signaling nodes in resistance to MAPK pathway inhibitors. However, they do not validate any of them. As an alternative to the previous Point 1 (ideally in addition), the authors should test if targeting some of the proposed signaling nodes reverses resistance to treatment.
3. If Ephrin and EGFR contribute to resistance, they may do so by activating the PI3K/AKT pathway, which acts downstream these RTKs, and which is often involved in resistance to MAPK pathway in inhibitors. Observing activation of the PI3K/AKT pathway would provide a positive control for the analysis. Do the authors observe an activation in their resistance model? Do PI3K and AKT inhibitors reverse resistance?

Minor points

4. As for the phosphoproteomics data, the authors quantified 3,207 phosphosites, which seems low for an experiment carried out with an Orbitrap Exploris. This suggests that the phosphoproteomic experiment did not use optimized methods.
5. Regarding the pMEK and pERK phosphorylation shown in Fig 1B, one can assume that these were measured by LC-MS/MS. However, there is no information in the Figure on the number of replicates in the experiment, the actual sites that were measured or which isoforms of MEK and ERK these sites belong to. This information should be provided.
6. There are several metrics of centrality that can be applied to network nodes. It is not clear which of them was applied to Fig 3A.
7. It is mentioned that ULK1-S556 shows strong upregulation by co-treatment. However, it is not clear if this is statistically significant as there only seem to be just one data point for the combination treatment in Fig 3C.
8. The authors write: "Although ARID1A-KO cells are resistant to MEK/BRAF inhibition (Fig. 4A),". However, Fig 4A does not support this statement.

Reviewer #3:

In their article, "ARID1A-induced transcriptional reprogramming rewires signalling responses to melanoma drug treatment", Petsalaki and colleagues describe a data set with intensive computational analysis that compares the transcriptome and proteome response of two isogenic cell lines (one with ARID1A KO, one control) to RAF and MEK inhibition. In an initial CRISPR screen, they identify ARID1A-loss as resistance mechanism. The main conclusions are several hypothesis about how ARID1A-loss leads to resistance, including JNK/JUN activation and increased RTK expression.

Overall the manuscript is well written, and can be well understood (despite its very detailed descriptive nature and excessive references to supplementary material). The extensive computational analysis of the multi-omics data set seems very well done and provides reasonable and interesting hypothesis of how resistance occurs. However, while these are interesting, it remains unclear why the authors did not test them.

As is, the manuscript remains largely descriptive and its implications speculative. Yet, the main hypotheses would be relatively easily testable: For instance, one could test if JNK or RTK inhibitors synergize with MEK/BRAF inhibitors in the ARID1A KO and not (or less) in ARID1A WT. While those synergies are not completely novel (see e.g. PMID 29431699, 23752269, <https://doi.org/10.1101/2024.05.13.593849> for RTK and JNK with MEK in CRC), this would strengthen the manuscript and add mechanistic insights into this mode of resistance and link it to ARID1A for Melanoma.

Detailed comments:

- Some of the figure legends and/or axis labeling are very vague, such that one does not really understand what is plotted. For instance: In figure 1A/B the legend describes the overall message, but not what is actually plotted and in figure 1A the axis label is not clear (LFC of what?).
- The initial CRISPR screen should be a bit more clearly described in the results (how long, "two additional screens"="two replicates"?). A rationale why MEK inhibitors were used in the screen would be interesting (and why not a RAF).
- SPRY1 etc are not phosphatases, in contrast to the labeling in Fig. 2D
- Some of the references are wrongly displayed in the main manuscript (such as on line 177).
- Font sizes in the figures are in part too small making it very hard to read.
- Line 534: Please say how many peptides are actually used, and not how many are on a "typical" chip.

Reviewer #1:

In this work, Barker et al attempt to infer mechanisms of TKI resistance (Trametinib, Vemurafenib and combination of the two) in sensitive melanoma A375 and in resistant ARID1A-KO A375 cell lines. They performed multi-modal characterisation (transcriptomics, proteomics and phosphoproteomics) and performed MOFA analysis to identify three key emerging factors: Factors 1 and 2 capture adaptive responses to drug treatments whereas Factor 3 reflects sustained resistance due to ARID1A loss. On top of this performed complex network analysis using the Phuego Tool, developed in the group, to identify topological network structures involved in individual processes, for each key factor they provided biological conclusions. In particular, authors hypothesize in resistant ARID1A-KO A375 cell lines treated with TKI a reactivation of the MAPK pathway by the EGFR and ROS1 receptors converging on JUN via FYN and PRKD1. Also, they propose a role of ARID1A-mediated resistance in suppressing immunogenicity of melanoma cancer cells, with clear critical implication for immunotherapy in this type of cancer.

The central conclusions are generally supported by a good and body of analyses and the study contributes to shedding light on resistance mechanisms in case of ARID1A loss (which constitute almost 10% of melanoma cases). For these reasons, I sincerely believe that the study will appeal to systems biologists and I support the publication.

However, I have the impression that the manuscript presentation lacks clarity and require a revision of text and figures: the many types of comparisons (ARID1A-KO vs parental, Vemurafenib-ctrl, Trametinib -ctrl, Combination-ctrl, treated ARID1A-KO vs trated parental., etc...) really hampers the readability and the clarity of the manuscript, in some instances, the interpretation could be better contextualized or clarified, the presentation of comparisons throughout the figures is at times unclear or incomplete, which hinders interpretability. Some conclusions-such as the suppression of immunogenicity-would benefit from more direct supporting evidence or elaboration and perhaps in vitro validation. Finally, the translational significance could be further highlighted, especially in terms of clinical relevance or therapeutic implications.

MAJOR REVISIONS:

1.1 The title of the work is ARID1A-induced transcriptional reprogramming rewires signalling responses to drug treatment in melanoma, however we end up in this only in Figure 4, despite I believe that also the first part is interesting, authors should clarify what is the relevance to the present manuscript of the first paragraphs.

R1.1 We thank the reviewer for the observation. To better reflect the content of the manuscript, including our insights into the mechanisms of drug response we have now changed the title to: **“Integrative multi-omics defines melanoma drug response networks and ARID1A-dependent resistance mechanisms”**

1.2 The paragraph "The basal transcriptional state of cells following ARID1A KO influences response to drug treatment" looks to me confusing and I struggled in following text statements in

the figures. Also, I am missing the point... If this cells are strongly resistant, why should they display a signalling rewiring after treatment? Also, did the author check whether drugs effectively are capable to enter ARID1A KO cells? This can in principle explain why there is no rewiring of the signalling. IHave

R1.2 We thank the reviewer for this valuable comment. We agree that the rationale of this section needed clarification and have now rewritten the paragraph accordingly.

Our data show that ARID1A loss reprograms the basal transcriptional and proteomic landscape *before* drug exposure, resulting in a distinct network topology that determines how signals propagate once the drug perturbs the system. Upon MEK/BRAF inhibition, both parental and ARID1A-KO cells display strong proteomic and transcriptomic responses (captured by Factor 1; Fig. 1D), demonstrating that the drug acts effectively on both lines and that there is no impairment in drug uptake.

However, because ARID1A-KO cells start from a rewired baseline state(Factor 3; Fig. EV7A–B), the signaling consequences of the same perturbation differ: phosphoproteomics and kinase-activity profiling reveal persistent ERK signaling, increased EGFR activity and expression at the membrane, suppressed FYN and PRKD1 activity and increased Ephrin-receptor expression among several other changes (Fig. 4A, C, G). Thus, resistance arises not from failure of the drug to engage its targets but from adaptive rerouting of signaling downstream of the drug perturbation, which maintains survival signaling in the ARID1A-KO.

1.3 LACK OF CLARITY: There are many comparison groups, it's difficult to follow which comparison is displayed/discussed, I strongly encourage the authors to make an extra effort in text and especially in Figures, where often this info is missing.

- I suggest the author to consistently refer in text and figure to ARID1A KO cells, as ARID1A-KO (some time it's just ARID1A (see as an example EV1) which is confusing).*
- It's difficult for a non-expert of some of the analyses and tools to follow some parts. For example, it's unclear to me in Fig. 1C (Phuego network) what up/down mean...*
- Similarly, in Network in Figure EV5, what are the up- or down-regulated processes? It's a Phuego output? It's the proteins' FoldChange? In case Fc in which condition or dataset?*
- Overall, the work jump from drug-control to ARID1A-KO - parental analyses, which makes the readability difficult, can the work be divided in clear distinct paragraph?*
- Lines 276-278, about the statement "their transcriptomic and proteomic drug response networks remain similar to parental cells (Fig. 1D; Fig. EV7A, EV7B)", I cannot find, in mentioned figures, the reported information. Which comparisons are you considering? ARID1A-KO vs parental or ARID1A-KO(treated) vs parental(treated)?*

R1.3 We apologise for the lack of clarity and have made effort throughout the text and figures to use consistent naming of our cell lines (ARID1A-KO & ARID1A-WT) and be explicit about all comparisons mentioned.

To help interpretation of the up and down-regulated phuEGO networks (Fig1D, 2C, 3B, (previous EV5A, EV6A, EV7 A & B; current Appendix Figure S2, 3 and EV4 A & B)) we have also added a clarification at the end of the first results section: "The resulting positive- and negative-weight networks represent signaling modules that are respectively upregulated or

downregulated in association with each factor, thereby capturing opposing functional programs linked to drug response or ARID1A loss.”

We also went throughout the text and tried to clarify the rationale for our analyses and separate the analyses on the ARID1A-WT vs KO cells. Where this is not possible we tried to improve readability and clarity in the paragraph. For example we can't separate the analyses in the paragraph on the basal network rewiring of ARID1A-KO cells, as the entire section describes how we integrate the information from the two conditions and networks to describe the network rewiring. Nonetheless we have rewritten the section to enhance clarity.

Regarding the last point the sentence we apologise for the omission of the relevant plot. We have added Supplementary Figure EV4C, which show a high correlation between the transcriptomic ($R = 0.91-0.93$, $p < 2.2e-16$) and proteomic responses ($R = 0.7-0.8$, $p < 2.2e-16$) in the ARID1A-KO vs the ARID1A-WT cells.

OTHER

1.4 At the phosphoproteomic level are direct substrate of BRAF or MEK changing? When checking for this I noticed that the phosphoproteomic dataset ST3 has only 900 rows, where only one displays a significant adj-pvalue, in text I read that there should be more, please check the table is correct.

R1.4 We thank the reviewer for this careful observation. The table initially displayed only the subset of phosphosites that were retained after preprocessing for MOFA analysis, rather than the full phosphoproteomic dataset used for differential abundance calculations. This has now been corrected. The updated supplementary tables include the complete set of quantified phosphopeptides (~3,000 across experimental conditions; **Table EV2**), and all log fold changes (LFCs) were recalculated using the full dataset to avoid potential bias (i.e., statistical double-dipping). We confirm that this correction does not affect any downstream analyses or conclusions, as differential abundance testing was performed independently of the MOFA or network inference steps.

Regarding the reviewer's question on BRAF and MEK substrates: both kinases have relatively few direct phosphosite targets, and coverage of these in our dataset was limited due to the inherent sparsity of phosphoproteomic measurements. For this reason, we complemented our analysis with kinase activity profiling assays (PamGene kinomics) and Luminex validation which show clear inhibition of these kinases. In contrast, ERK, a more promiscuous downstream kinase, showed broader representation in our phosphoproteomic data. Using phosX (see Figure embedded in this response), we observed a significant reduction in phosphorylation of known ERK substrates relative to the background, consistent with effective inhibition of the MAPK pathway. These findings are in agreement with our kinase activity and Luminex data (**Fig. 1B**, **Fig. EV1D**), confirming that BRAF/MEK inhibition was achieved at both the signaling and functional levels.

Kinase activity volcano plot of A375 melanoma cells following combination therapy. Each point represents a kinase, with the x-axis showing the activity score (log fold change) and the y-axis representing the $-\log_{10}$ adjusted p-value. Significantly upregulated kinases are shown in red, downregulated kinases in blue, and non-significant kinases in grey. Key MAPK pathway kinases (MAPK1, MAP2K1, and BRAF) are highlighted in blue and labeled. Dashed lines indicate the thresholds for statistical significance ($p < 0.05$) and activity score cutoffs ($|\logFC| > 0.2$).

Given the limited direct coverage of BRAF/MAP2K1 substrates in the phosphoproteomic dataset and the consistent evidence from independent kinase activity and Luminex assays, we chose not to include these sparse phosphosite-level results in the main text to maintain focus on the more robust findings.

1.5 In Table S2-4, Authors should add also statistics for **all** the comparisons included in the study.

R1.5 We apologise for the error. We have now included the LFC and associated statistics for all comparisons represented in our data (drug conditions vs no drug, ARID1A-WT vs ARID1A-KO etc). These can be found in the updated **Table EV2-4**.

1.6 Fig. 2C is rich of information, however is not discussed in text. About Fig.2C, FC in abundance to which comparison refers to? All treatment vs untreated? Please add info... (Similarly, add info to and Fig. 3B.) Also, what does 'semantic similarity' indicate? Is it relevant?

R1.6 We apologise for the lack of clarity in the figures. We have updated the respective results sections to discuss the presented phuEGO networks, where the information was missing, and to link our observations with the rest of the data presented in the figure.

Semantic similarity indicates that two nodes are functionally similar. For example phuEGO uses the semantic similarity between edges to preferentially propagate signals towards nodes that are more likely to work together compared to functionally distinct ones. We have therefore included it in the network to highlight potential regions of the network that have more functionally similar densely interconnected regions. We added an explanation for the term in the legend of the figure.

1.7 Line 231, about "Despite phosphoproteomics being the main driver for this factor", where did you get this info from?

R1.7 We apologise for the omission. The % of variance in each data modality explained by each factor is shown in Figure EV3 (previously EV4). We now refer to this figure where we make the statement about phosphoproteomics being the main driver for this factor.

1.8 Lines 233-236 "Looking at the phosphosites that drive the variance captured by Factor 2, we find several phosphoproteins involved in DNA repair-related functions (Fig. 3C and Table S5)." How did the authors find this information? Can they quantify? which are these DNA-repair proteins (only the ones in Fig. 3C)? Are there other processes? .

R1.8 We apologise for the lack of clarity. These are phosphosites that are present on proteins that are in the network derived from phuEGO. They are also significantly differentially abundant. We have explained this in the legend of Figure 3C. The function annotation was done based on literature. We have now revised the figure to include the functional annotations of the proteins. Additional functions captured by Factor 2 are shown in Fig 3D and Appendix Figure S3D (previous Fig EV6D).

1.9 Lines 298-300, authors should exclude that there is HUBS bias here, I suggest to do a randomisation.

R1.9 We thank the reviewer for raising this important concern regarding potential hub bias. We would like to clarify that the *phuego* framework explicitly incorporates mechanisms to address this issue.

As detailed in the *phuego* publication (Giudice et al., 2024), the algorithm mitigates hub bias in several ways:

1. The method includes a benchmarking step in which 1000 random networks are generated preserving the node degree distribution. This ensures that the enrichment

results from the propagated network are not simply due to structural properties like node centrality.

2. To further correct for hub-related biases, *phuego* applies Laplacian normalization, whereby each edge weight is scaled by the square root of the weighted degree of the interacting nodes. This dampens the influence of highly connected hubs and balances the contribution of lower-degree nodes.

Given these in-built controls, we are confident that the observed signal enrichment and network prioritizations in our study are not driven by hub artifacts. For transparency, we have now added this explanation to the Methods section of the manuscript “Network Propagation using *phuego*”.

1.10 The model in Figure 4F is very nice, could it be integrated with literature info, to add directionality to the edges, since we are describing a signalling network. I think this could really add information.

R1.10 We thank the reviewer for the suggestion. We have amended Fig 4F to integrate regulatory information (activation/inhibition) into the network.

1.11 To conclude, have authors considered to perform in vitro validation of this results?

R1.11 We thank the reviewer for the suggestion. Extensive in vitro validation of our network analysis is not feasible within a reasonable timeframe, as it requires the generation, validation, and maintenance of multiple KO cell lines, access to multiple machines that can measure cell growth over time in the presence and absence of treatment. As such, it is therefore out of scope for this manuscript. We have nonetheless set up an imaging based killing assay and performed a pilot proof-of-principle with such an experiment to show that indeed EGFR is a main driver of drug resistant signalling: targeting *EGFR* completely abolishes drug resistance in the ARID1A-KO cell line and induces rapid cell death. This data is now presented in 4H.

We would also like to point out that we have begun exploring whether ARID1A loss influences immune-mediated killing. In preliminary experiments using a bispecific antibody system, ARID1A-KO cells appeared to be killed more slowly than parental cells, suggesting a possible immune-related phenotype. However, as bispecific molecule engaging CD3- mediated killing does not fully recapitulate physiological T-cell recognition, a more detailed evaluation using engineered TCR-based models will be required. These experiments are ongoing but beyond the current scope of this study, and we plan to investigate this mechanism further in future work.

MINOR POINTS:

1.12 Referring to lines 132-135, I have the impression that EV3B refers to the entire dataset and not to the significant ones.

R1.12 This is indeed true and it is mentioned in the figure legend; “*Upset plot showing the extent of distinct proteins, genes and peptides quantified from each 'omics experiment. Barplots*

show the number of readings counted in each intersection of the data.” The figure is meant to give a ‘big picture description of our multiomics data, both considering the differentially abundant entities and overall. Please note that the previous EV3B is now EV2B.

1.13 About the "correlates strongly" in line 189, I acknowledge that the R tend to be good, however, I would reduce the statement as it's significant in only one case (EV5D) .

R1.13 We have changed ‘strongly’ to ‘well’. However we would like to point out, that given how few data points are considered here, and the fact that we are comparing completely independently measured kinase activity with network centrality extracted from networks inferred from a factor analysis, the fact that there is any correlation at all provides strong support for the relevance of our networks. We would also argue that a significance cutoff of 0.05 p-value is rather arbitrary and one can easily set it to ~0.1 making all but 1 condition ‘non significant’.

1.14 Line 230, please replace MAPK2K1 with MEK, as you used this nomenclature in the rest of the manuscript.

R1.14 This has been amended

Reviewer #2:

Summary

In this manuscript, Barker et al used graph methodology to investigate mechanisms of resistance to MAPK pathway inhibitors in a ARID1A-KO melanoma cell line relative to its WT counterpart. To this end, the authors integrate transcriptomic, proteomic and phosphoproteomics data using innovative approaches, some of which were developed by the Petsalaki group. The authors used multiomic factor analysis to identify three "factors" that drive variation across the RNA-seq, proteomics and phosphoproteomics data. The first two factors capture adaptive response to treatments while Factor 3 capture resistance due to ARID1A loss. The loadings from MOFA were then used as input of a network propagation method named phuEGO, which was recently developed by the group. The authors conclude the elevated RTKs such as EGFR and ROS1 and the downstream signaling nodes PRKD1, JUN and NCK1 contribute to resistance.

General remarks

The study seems to be well conducted from the computational standpoint and the authors place emphasis on the use of innovative approaches to multiomic data analysis.

2.0 It is not clear, however, whether such new methods (phueGO, MOFA) add insights that could not have been obtained with the use of standard means to analyze omics data. I think showing this is of key importance for the paper to have novelty from the methodological standpoint. On the other hand, if the authors claim that the study is not a methods paper, and instead it shows new biological insights, then more work should be done to validate such new

knowledge. In summary, the impression of this reviewer, from the way that the manuscript is presented, is that the study falls short of having methodological or validated biological novelty.

R2.0 We thank the reviewer for this thoughtful and constructive comment. We respectfully disagree that our analysis lacks novelty, and we have revised the manuscript to clarify both the methodological innovation and the biological insights it enabled.

Our approach is not intended as a methods paper, but it introduces an analytical framework that extends beyond standard omics workflows. Specifically, we integrate multi-omics factor analysis (MOFA), which decomposes molecular data into orthogonal latent sources of biological variation, with our network-based phuEGO method, which embeds these factors into context-specific signaling networks. This combination allows us to link statistically independent molecular programs to their mechanistic signaling underpinnings, revealing interactions between layers of regulation that are not accessible through standard single-omics or enrichment analyses.

This framework provides three key advantages:

1. **Improved interpretation and hypothesis generation:** it decomposes complex datasets into distinct, biologically meaningful processes (e.g., drug response vs. ARID1A-dependent reprogramming).
2. **Mechanistic integration across data layers:** it contextualizes molecular changes across the transcriptome, proteome, and phosphoproteome, enabling the identification of upstream–downstream relationships such as the EGFR–ROS1–JUN axis mediated by PRKD1, FYN, and PTPN6.
3. **Experimental prioritization:** it identifies key nodes whose regulation would not have been apparent from any single omics layer. For example, while EGFR appeared downregulated at the total proteome level, our integrated network analysis predicted a functional role for EGFR signaling in ARID1A-mediated drug resistance. This prompted us to examine EGFR surface localization experimentally, revealing a marked increase in membrane EGFR in ARID1A-KO cells, findings consistent with the proposed mechanism.

Furthermore, this analysis revealed that drug-induced variance (Factor 1) is shared across treatments and genotypes, whereas Factor 3 captures the ARID1A-dependent basal reprogramming that modifies signaling responses to therapy, an insight not achievable with standard differential or enrichment analyses.

To strengthen the biological conclusions, we have now included experimental validation of the predicted role of EGFR as an essential node in ARID1A-KO drug resistance (Fig. 4H). These validations confirm the predictive power and biological relevance of our integrative analytical approach.

We have added these clarifications to the revised Results and Discussion sections.

Major points

2.1. As pointed above, the authors should show that complex graph-theoretical methodology offers advantages for the analysis of omics data. What information has this analysis provided that could not have been obtained from a standard analysis of the data?

R2.1 We thank the reviewer for this important point. The aim of the graph-theoretical analysis in this study is to mechanistically integrate the orthogonal sources of variation captured by the 'omics analysis (represented by MOFA) using prior knowledge. While both MOFA and network propagation have been previously described in the literature, we agree that the specific added value of combining these approaches could be more explicitly explained in our manuscript.

To illustrate this, we now provide inline with this response some additional analysis showing a direct comparison between standard enrichment analysis on genes with high factor loadings from MOFA and the enrichment results obtained after applying the phuego network propagation algorithm. Specifically, we compared the GO Biological Process terms enriched in MOFA-derived gene lists to those enriched in network-propagated gene sets for each factor. Given that the paper itself is not a methods paper, we have left this to the rebuttal and not the main text of the paper.

Our results show that the network propagation step consistently increases the fold enrichment of many biological processes (Figure below). This demonstrates the signal-amplifying effect of network-based integration by leveraging prior biological knowledge to reinforce coherent pathways that may be only weakly present in the original multi-omics data. This is a key feature of the phuEGO algorithm described in the original paper, although it has not previously been integrated with MOFA to mechanistically understand direct outputs from this analysis.

Furthermore, by comparing the direct rank of enriched terms before and after network propagation, we show that phuEGO selectively boosts signals associated with biological mechanisms relevant to melanoma and drug resistance. For example, in both Factor 1 and Factor 3, terms related to regulation of phosphorylation, response to growth factors, and MAPK signaling are significantly up-ranked after network analysis. These are processes highly relevant to kinase-driven resistance phenotypes. Notably, central nodes such as MAPK1 and MAPK3 are not among the top-ranked genes in the MOFA factor loadings alone (falling into the 14th and 56th percentile in Factor3, respectively), yet emerge as key components in the network due to their connectivity and propagation behavior. These same kinases were experimentally validated to have altered activity in our kinase assays, further reinforcing the utility of the network-based signal refinement.

Finally, network analysis allows us to move beyond gene list interpretation by providing a mechanistic framework that links genes through functional interactions. This enables us to better contextualize the biological pathways involved and explain the interplay of molecular components underlying resistance phenotypes. This is a major advantage over traditional 'omics analysis alone.

We hope this analysis clarifies the added value of our network-based approach and demonstrates how it strengthens both the interpretability and biological relevance of the multi-omics data. We have also added text in the discussion to reinforce the added benefit as described in this response. (see Lines 502–517).

Comparison of GO term enrichment between MOFA factor loadings and network propagation. For each MOFA factor (Factor1–Factor3), gene sets derived from high or low factor loadings were analyzed for GO Biological Process (BP) enrichment using standard methods (MOFA output) and after applying the phuEGO network propagation algorithm. Panels A–C (scatter plots): Fold enrichment (OR) of GO terms from network propagation (x-axis) versus MOFA output (y-axis) for each factor. Points are colored according to significance: red (both significant), blue (factor significant only), green (network propagation significant only), and gray (neither significant). The dashed line indicates equality. Panels D–F (rank plots): Change in rank of GO terms between network propagation and MOFA-only analysis. Points represent individual GO terms, with lines connecting them to the baseline ($x = 0$). Terms with the largest positive or negative changes are labeled. The x-axis shows rank change (network propagation - MOFA), and the y-axis shows the ranked GO terms (labels hidden for clarity). Positive delta values indicate terms up-ranked after network propagation. These plots illustrate the added value of network propagation in amplifying biologically coherent signals that may be underrepresented in MOFA factor loadings alone, highlighting key processes involved in melanoma drug resistance, such as MAPK signaling, regulation of phosphorylation, and response to growth factors.

2.2. The major issue, aside from the point above, is that the authors propose the involvement of several signaling nodes in resistance to MAPK pathway inhibitors. However, they do not validate any of them. As an alternative to the previous Point 1 (ideally in addition), the authors should test if targeting some of the proposed signaling nodes reverses resistance to treatment.

R2.2. Extensive in vitro validation of our network analysis is not feasible within a reasonable timeframe, as it requires the generation and maintenance of multiple KO cell lines, access to multiple machines that can measure cell growth over time in the presence and absence of

treatment. As such, it is therefore out of scope for this manuscript. We have nonetheless done a proof-of-principle with such an experiment to show that indeed EGFR is a main driver of drug resistant signalling: knocking out EGFR completely abolishes drug resistance in the ARID1A-KO cell line and induces rapid cell death. This data is now presented in Fig 4H.

2.3 If Ephrin and EGFR contribute to resistance, they may do so by activating the PI3K/AKT pathway, which acts downstream these RTKs, and which is often involved in resistance to MAPK pathway in inhibitors. Observing activation of the PI3K/AKT pathway would provide a positive control for the analysis. Do the authors observe an activation in their resistance model? Do PI3K and AKT inhibitors reverse resistance?

R2.3 We thank the reviewer for this insightful comment. Indeed, activation of the PI3K/AKT axis is a well-established mechanism of adaptive resistance to MAPK pathway inhibition. However, in our data, we do not observe substantial activation of this pathway. Specifically, functional kinomics (PamGene) data for AKT1 and AKT2 show no significant change in activity following drug treatment in either ARID1A-WT or ARID1A-KO cells. Our phosphoproteomic data do not contain confidently quantified PI3K/AKT pathway phosphosites.

These findings suggest that ARID1A-mediated resistance proceeds through an alternative signaling route, primarily involving EGFR–ROS1–FYN–PRKD1–JUN interactions, as indicated by our integrated network analysis and validated experimentally (Fig. 4). Interestingly PI3K/AKT do show up in our upregulated network related to the combination treatment (Fig 3B), but as this pathway is not supported by directly relevant measured data points, but has rather been introduced through the network analysis, we would rather not present it as one of the mechanisms of cell adaptation to treatment.

While we did not test PI3K or AKT inhibitors directly in this study, the absence of AKT activation together with strong rewiring through alternative RTK-dependent pathways indicates that PI3K/AKT signaling does not play a dominant role in this resistance model.

Minor points

2.4. As for the phosphoproteomics data, the authors quantified 3,207 phosphosites, which seems low for an experiment carried out with an Orbitrap Exploris. This suggests that the phosphoproteomic experiment did not use optimized methods.

R2.4 We thank the reviewer for this observation. The phosphoproteomics experiments were conducted as part of the European EPIC-XS initiative (<https://cordis.europa.eu/project/id/823839>) in collaboration with a leading expert laboratory, ensuring high technical quality. Across five TMT runs, we identified on average 8,838 phosphosites per run, consistent with high-quality Orbitrap Exploris datasets. The final number of 3,207 quantified sites reflects stringent filtering and integration across runs to ensure reproducibility and comparability with the corresponding proteome and transcriptome datasets.

The relatively lower coverage is attributable to TMT batch design, which was originally optimized to capture differences between ARID1A-KO and WT cells rather than drug-induced perturbations, now known to drive the strongest phosphoproteomic changes. While the cost of repeating the experiment with new batch composition is prohibitive, our multi-omics integration and network-based analysis effectively leveraged this dataset to extract robust mechanistic insights. In addition, we complemented phosphoproteomic results with functional kinase activity profiling (PamGene), which independently validated the key signaling nodes identified in our analysis.

2.5. Regarding the pMEK and pERK phosphorylation shown in Fig 1B, one can assume that these were measured by LC-MS/MS. However, there is no information in the Figure on the number of replicates in the experiment, the actual sites that were measured or which isoforms of MEK and ERK these sites belong to. This information should be provided.

R2.5 The data shown in Figure 1B were obtained using the Luminex assay, as described in the Methods section, rather than by LC-MS/MS. The experiment used two independent gRNAs targeting the same gene, providing biological replication through distinct perturbations rather than technical repeats. As established in Figure EV1D, the Luminex assay exhibits minimal technical variability (see figure below); therefore, single measurements were performed for each gRNA. This clarification has been added to the Figure 1B legend.

Technical reproducibility of the Luminex assay. Pairwise correlations between triplicate wells across all conditions were consistently high (median $r > 0.99$). Based on this reproducibility, subsequent measurements were performed as single wells per condition.

2.6. There are several metrics of centrality that can be applied to network nodes. It is not clear which of them was applied to Fig 3A.

R2.6 We apologise for the lack of clarity. The network in Fig. 3A uses PageRank centrality (see y axis label) to quantify node importance. This metric captures both direct and indirect influence within the network by weighting nodes according to the connectivity and importance of their neighbors, which we found well suited for identifying signaling hubs in our diffusion-based networks. We have clarified this in the figure legend of the revised manuscript.

2.7. It is mentioned that ULK1-S556 shows strong upregulation by co-treatment. However, it is not clear if this is statistically significant as there only seem to be just one data point for the combination treatment in Fig 3C.

R2.7 We apologise for the omission. We have now amended Fig 3C to add significance metrics and explain in the legend the statistical test and the number of replicates (n=3 for each condition).

2.8. The authors write: "Although ARID1A-KO cells are resistant to MEK/BRAF inhibition (Fig. 4A)". However, Fig 4A does not support this statement.

R2.8 We apologise for the confusion. Fig 4A shows that upon drug treatment ARID1A-KO cells don't show suppression of MAPK or JNK activities (first line of heatmap) which is the typical result of BRAF/MEK inhibition and is robustly happening in the ARID1A-WT cells (second line of heatmap). We have amended the labels of the heatmap to make the comparison clearer.

Reviewer #3:

In their article, "ARID1A-induced transcriptional reprogramming rewires signalling responses to melanoma drug treatment", Petsalaki and colleagues describe a data set with intensive computational analysis that compares the transcriptome and proteome response of two isogenic cell lines (one with ARID1A KO, one control) to RAF and MEK inhibition. In an initial CRISPR screen, they identify ARID1A-loss as resistance mechanism. The main conclusions are several hypothesis about how ARID1A-loss leads to resistance, including JNK/JUN activation and increased RTK expression.

3.1 Overall the manuscript is well written, and can be well understood (despite its very detailed descriptive nature and excessive references to supplementary material). The extensive computational analysis of the multi-omics data set seems very well done and provides reasonable and interesting hypothesis of how resistance occurs. However, while these are interesting, it remains unclear why the authors did not test them.

As is, the manuscript remains largely descriptive and its implications speculative. Yet, the main hypotheses would be relatively easily testable: For instance, one could test if JNK or RTK inhibitors synergize with MEK/BRAF inhibitors in the ARID1A KO and not (or less) in ARID1A WT. While those synergies are not completely novel (see e.g. PMID 29431699, 23752269, <https://doi.org/10.1101/2024.05.13.593849> for RTK and JNK with MEK in CRC), this would

strengthen the manuscript and add mechanistic insights into this mode of resistance and link it to ARID1A for Melanoma.

R3.1 We thank the reviewer for the suggestion. In the revised manuscript we have experimentally validated the central role of EGFR in mediating drug resistance in the ARID1A-KO melanoma cells.

Detailed comments:

3.2 Some of the figure legends and/or axis labeling are very vague, such that one does not really understand what is plotted. For instance: In figure 1A/B the legend describes the overall message, but not what is actually plotted and in figure 1A the axis label is not clear (LFC of what?).

R3.2 We apologise for the lack of clarity. We have now gone over the figures and made the comparisons and labels clearer.

3.3 The initial CRISPR screen should be a bit more clearly described in the results (how long, "two additional screens"="two replicates"?). A rational why MEK inhibitors were used in the screen would be interesting (and why not a RAF).

We conducted two replicate genome-wide CRISPR screens using Cas9-expressing A375 cell lines. These screens were performed with Trametinib, since two previous studies had already carried out genome-wide screens using 2 uM Vemurafenib on A375 cell lines (Shalem et al, 2014 (<https://orcs.thebiogrid.org/Screen/95>), Doench et al 2016 (<https://orcs.thebiogrid.org/Screen/584>)) (a BRAF inhibitor). The Trametinib screens were designed to determine whether the same genes would emerge as hits, indicating effects on the broader MAPK pathway rather than responses specific to a single drug target. The number of cells and the timings used for the screens are detailed in the method section. We have explained the rationale for using trametinib in text (line 112-117). "To assess whether ARID1A acts broadly across the MAPK pathway rather than mediating resistance specific to a single drug, we conducted two additional screens in A375 cells using a different MEK inhibitor, trametinib. These screens again identified *ARID1A*, along with established resistance genes (e.g., *NF1/2*, *KIRREL*, *MED12*, *TAF5/6L*), further supporting its role in BRAFV600E melanoma resistance."

3.4 SPRY1 etc are not phosphatases, in contrast to the labeling in Fig. 2D

R3.4 We thank the reviewer for the observation. We have amended the label to read 'negative feedback regulators'.

3.5 Some of the references are wrongly displayed in the main manuscript (such as on line 177).

R3.5 This has been amended.

3.6 Font sizes in the figures are in part too small making it very hard to read.

R3.6 We have updated the figures and increased the fonts where possible

3.7 Line 534: Please say how many peptides are actually used, and not how many are on a "typical" chip.

R3.7 This has been amended. The information can be found in the 'Activity profiling through functional kinomics' section in the methods.

5th Dec 2025

Manuscript Number: MSB-2025-12988R

Title: Integrative multi-omics defines melanoma drug response and ARID1A-dependent resistance mechanisms

Author: Charlie Barker

Sumana Sharma

Ana Mafalda Santos

Konstantinos-Stylianos Nikolakopoulos

Athanasios Velentzas

Cristina Tormo-Garcia

Anushka Sharma

Franziska Völlmy

Angeliki Minia

Vicky Pliaka

Joseph Clarke

Maarten Altelaar

Gavin Wright

Leonidas Alexopoulos

Dimitrios J. Stravopodis

Evangelia Petsalaki

Dear Evangelia,

Thank you for sending us your revised manuscript. We have now heard back from the three reviewers who were asked to re-evaluate your study. As you will see, the reviewers are all satisfied with the modifications made. Before we can formally accept your manuscript, we would ask you to address the following editorial-level issues:

1. Please include up to five keywords in the manuscript file.
2. Remove "Authors' contribution" section from the manuscript file.
3. The Wellcome Trust grant number 317519/Z/24/Z is missing from the "Acknowledgment" section. Please add it.
4. Data availability:
 - "Data and code availability" should be renamed to "Data availability".
 - Please provide specific URLs for the datasets in the data availability statement: PRIDE (identifier: PDX058857) and gene expression data (identifier: E-MTAB-14759)
5. There are callouts for Tables S1-S4, but we are unable to locate them. Please double-check. Additionally, please use consistent nomenclature such as EV tables/datasets, or refer to them as "Appendix Table Sx".
6. EV tables and datasets :
 - Update all source file names, titles, legends, and manuscript callouts to refer to Dataset EV1-EV# instead of Table EV1-EV6. Remove the legends from the manuscript file.
 - Table EV7-EV8 should be renamed to Table EV1-EV2, with the corresponding callouts and legends removed from the manuscript and placed above each table in the respective Excel file.
 - Data S1 is not permitted. Please remove the file and its callout in the text. If necessary, it can be provided as source data.
7. Source data:
 - Source data for main figures should be uploaded as one (zipped) file /figure, and named as "manuscriptID_SourceDataForFigure x".
 - Source data for supplementary figures should be combined in one file.
 - Please provide a completed Source data checklist (sent to you on May 19th) and upload it as "Related Manuscript File".
8. Please address the following issues in figure legends :
 - Please note that the exact p values are not provided in the legends of figures 3c; EV-4c;Appendix Figure S1-c.
 - Please indicate the statistical test used for data analysis in the legends of figures 3d; 4e,h; 5a; EV-4c.
 - Please note that the box plots need to be defined in terms of minima, maxima, centre, bounds of box and whiskers, and percentile in the legends of figures Appendix Figure S2-b; Appendix Figure S4-c.
 - Please note that scale bar and its definition are missing for figures EV-1c.

9. Sections need to be named and the order should be corrected: Title page - Abstract - Keywords - Introduction - Results - Discussion - Methods - Data Availability - Acknowledgements - Disclosure and Competing Interests Statement - References - Figure Legends - Table(s) - Expanded View Figure Legends.

Please use the following link to submit your revised paper:

Thank you for submitting this interesting paper to Molecular Systems Biology.

Kind regards,

Jingyi

Jingyi Hou, PhD
Senior Editor
Molecular Systems Biology

If you do choose to resubmit, please click on the link below to submit the revision online before 4th Jan 2026.

*** PLEASE NOTE *** As part of the EMBO Press transparent editorial process initiative (see our Editorial at <https://dx.doi.org/10.1038/msb.2010.72> , Molecular Systems Biology will publish online a Review Process File to accompany accepted manuscripts. When preparing your letter of response, please be aware that in the event of acceptance, your cover letter/point-by-point document will be included as part of this File, which will be available to the scientific community. More information about this initiative is available in our Instructions to Authors. If you have any questions about this initiative, please contact the editorial office (msb@embo.org).

Reviewer #1:

The authors addressed all the points raised.

Reviewer #2:

The authors have adequately addressed all the points I made in the previous review.

Reviewer #3:

The authors addressed all of my concerns (including a new validation experiment).

The authors have addressed all minor editorial requests.

10th Dec 2025

Manuscript number: MSB-2025-12988RR

Title: Integrative multi-omics defines melanoma drug response and ARID1A-dependent resistance mechanisms

Dear Evangelia,

Thank you again for sending us your revised manuscript. We are now satisfied with the modifications made and I am pleased to inform you that your paper has been accepted for publication.

You may qualify for financial assistance for your publication charges - either via a Springer Nature fully open access agreement or an EMBO initiative. Check your eligibility: <https://link.springer.com/journal/44320/how-to-publish-with-us>

Yours sincerely,
Jingyi

Jingyi Hou, PhD
Senior Editor
Molecular Systems Biology

>>> Please note that it is Molecular Systems Biology policy for the transcript of the editorial process (containing referee reports and your response letter) to be published as an online supplement to each paper. If you do NOT want this, you will need to inform the Editorial Office via email immediately. More information is available here: <https://link.springer.com/partners/embo-press/editorial-policies#Peer%20review>